# Conformal Mirror Statistics for Model Alignment: Uncertainty Quantification with FDR Control

## Abstract

Foundation models are increasingly adopted across diverse domains, but their safe deployment requires outputs that align with human interpretation, especially in high-stakes applications. This motivates the need for rigorous uncertainty quantification (UQ) methods to assess alignment reliability. Most existing methods rely on large labeled datasets, limiting their applicability in real-world settings where labeled data is scarce or expensive. In this paper, we introduce Conformal Mirror Statistics (CMS), a novel framework for UQ in model alignment, selecting aligned outputs for unlabeled data with the false discovery rate (FDR) under control. Unlike conventional conformal methods based on $p$-value calibration, CMS generalizes to broader settings without restrictive calibration size requirements. We further establish theoretical guarantees by proving FDR control under weaker data assumptions than existing methods. Empirical results on simulations and a large sepsis cohort from MIMIC-III demonstrate that CMS consistently outperforms conventional methods while reliably identifying aligned outputs.

## 1 Introduction

With the increasing adoption of deep learning and foundation models across diverse domains, these models are progressively taking over decision-making tasks, reducing human workload. However, due to inherent biases, distribution shifts, and other limitations, these models can produce misleading outputs (Gallegos et al., 2024; Oh et al., 2025; Ranjan et al., 2024). Unchecked reliance on these outputs can lead to significant consequences, particularly in high-stakes applications such as healthcare diagnostics, legal decision-making, and financial forecasting (Hager et al., 2024; Dahl et al., 2024; Chen et al., 2025). Ensuring that foundation model outputs align with human understanding has thus become a critical concern. This underscores the necessity of robust uncertainty quantification (UQ) techniques to assess and enhance the reliability of model alignment, mitigating potential risks associated with their deployment.

While classical UQ approaches such as confidence intervals and machine learning methods provide theoretical guarantees, they rely on strong modeling assumptions and do not extend to black-box predictors (Efron, 1981; Casella & Berger, 2002; Gelman et al., 2013). Conformal inference addresses these limitations by offering distribution-free guarantees for prediction-level reliability, and has been applied to multiple testing, outlier detection, and model alignment (Jin & Candès, 2023b; Gui et al., 2024). However, existing conformal approaches often require large labeled calibration sets, which are costly and time-consuming to obtain in practice. This limitation is particularly acute in domains such as medicine, where expert annotations are scarce and expensive, or in finance and law, where ground-truth labels may be ambiguous or delayed (Olatunji et al., 2019; Hendrycks et al., 2021; Choi et al., 2025). Moreover, their validity and power can degrade under complex data structures which are common in modern large-scale dataapplications.

To address these limitations, we propose *Conformal Mirror Statistics (CMS)*, a novel framework for uncertainty-aware model alignment with minimal labeled data. Figure 1 illustrates the workflow: given an unlabeled dataset, the CMS algorithm evaluates whether the outputs of a model (e.g., transformer) are reliable. The construction of CMS leverages both magnitude and rank information to achieve asymptotic symmetry for false discovery rate (FDR) control. This enables thresholds to be

Figure 1: Workflow of the Conformal Mirror Statistics (CMS) algorithm. (a) Unlabeled test dataset consisting of model outputs awaiting evaluation. (b) CMS procedure: a small labeled dataset is used to train an alignment predictor and construct mirror statistics. (c) Alignment results on the test outputs, indicating which predictions are deemed unreliable.

determined directly from statistic magnitudes, rather than through the BH procedure on conformal $p$-value orderings, thereby preserving statistical power even in the presence of limited labeled data. Additionally, the method accommodates complex data structures, including non-exchangeability and heteroskedasticity. Unlike the original mirror statistics in Dai et al. (2023b), which are constructed from regression coefficients and require additional restrictions on both the coefficients and the statistics, our approach exploits the properties of alignment scores to build a new form of mirror statistics. As a result, our method achieves FDR control under weaker assumptions.

Empirically, we demonstrate that CMS achieves reliable alignment selection on heteroskedastic simulation settings and a large-scale sepsis cohort from MIMIC-III, outperforming conventional conformal methods in terms of FDR control and stability. More broadly, the framework is applicable to certifying outputs across a wide range of real world tasks, making it well suited for the growing landscape of foundation model applications in high-stakes domains.

### 1.1 MAIN RESULTS

We summarize our main contributions and findings as follows:

- **Remain power on large unlabeled datasets:** Our statistics mitigates the loss of power in existing methods that occurs when the labeled calibration set is small relative to the unlabeled test set size.

- **Greater tolerance to complex data structures:** The proposed statistic maintains FDR control under more complex data structures such as non-exchangeable data than previous conformal approaches, as supported by theory and simulation.

- **Novel statistic with weaker assumptions:** We construct a new form of mirror statistics for conformal inference and prove FDR control under weaker conditions than those required in prior mirror statistics work (Dai et al., 2023b). This broadens the theoretical foundation of methods for conformalized selection.

- **Validated on pratical tasks:** We validate CMS on simulations and a large-scale MIMIC-III sepsis cohort, where it outperforms baseline methods in FDR control. More broadly, CMS provides a general framework for certifying reliable outputs across diverse machine learning applications.

## 2 RELATED LITERATURE

**Conformal prediction and conformal inference.** Conformal methods provide distribution-free guarantees and have become a powerful tool for predictive uncertainty quantification. Conformal prediction (Shafer & Vovk, 2008; Lei et al., 2018; Angelopoulos & Bates, 2021) generates prediction sets with guaranteed coverage. Conformal inference instead focuses on selecting reliable outputs using conformal $p$-values (Jin & Candès, 2023b;a; Liang et al., 2024; Gui et al., 2024). While

effective, these methods require large labeled calibration sets, which remain costly and limit practical adoption.

**FDR control and mirror statistics.** The Benjamini–Hochberg procedure (Benjamini & Hochberg, 1995) established FDR control for $p$-values, and knockoffs (Barber & Candès, 2015) extended this idea to more flexible settings. Recently, mirror statistics (Dai et al., 2023b) have emerged as a robust tool for selective inference, achieving exact FDR control via data splitting. Subsequent work has further generalized mirror statistics across diverse tasks (Tong et al., 2023; Dai et al., 2023a), highlighting their robustness and versatility.

## 3 METHODOLOGY

### 3.1 PROBLEM SETTINGS AND PRELIMINARY

Suppose we have a fixed pre-trained foundation model $f : \mathcal{X} \to \mathcal{Y}$ that maps an input prompt to an output. We further assume access to a labeled dataset $\mathcal{D}_l = \{(X_i, L_i)\}_{i=1}^n$, where $X_i \in \mathcal{X}$ denotes the input prompt and $L_i \in \mathcal{L}$ is a reference label used to evaluate alignment. For example, consider a pretrained model that generates treatments from electronic health record (EHR) data: here $X_i$ corresponds to a patient's EHR, $f(X_i)$ is the model-generated treatment plan, and $L_i$ is the expert-provided diagnosis.

To evaluate whether the foundation model output $f(X_i)$ aligns with the true label $L_i$, we introduce an alignment function $\varphi : \mathcal{Y} \times \mathcal{L} \to \mathbb{R}$. This function compares the generated output $f(X_i)$ against the reference label $L_i$ and returns an alignment score $A_i = \varphi(f(X_i), L_i)$. In the treatment plan generation example, $A_i$ could represent a similarity measure between the LLM-generated plan and the expert-authored plan. Throughout this paper, given a suitable choice of alignment function, we treat $A_i$ as the true alignment score. If $A_i \leq c$, the output $f(X_i)$ is regarded as misaligned with $L_i$; if $A_i > c$, it is considered aligned, where $c \in \mathbb{R}$ is a pre-specified threshold.

In addition to the labeled data, we also have an unlabeled test dataset $\mathcal{D}_{\text{test}} = \{X_{n+j}\}_{j=1}^m$. Note that for a unit data $X_{n+j}$ without reference label information, the true alignment score of its generated output is unknown, and we do not seek to evaluate it, which often requires expert annotation or human judgment. Instead, the goal of this paper is to select a subset $\widehat{S} \subseteq [m] := \{1, \ldots, m\}$ such that most of their (unobserved) true alignment scores are above the pre-fixed threshold $c \in \mathbb{R}$, with the FDR under control. Let $S = \{j \in [m] : A_{n+j} > c\}$ be the true alignment set. Formally, we measure reliability by the false discovery proportion (FDP) and its expectation, the FDR:

$$\text{FDP} = \frac{\sum_{j \in [m]} I\left\{A_{n+j} \leq c, j \in \widehat{S}\right\}}{\max(|\widehat{S}|, 1)} \quad \text{with} \quad \text{FDR} = \mathbb{E}(\text{FDP}),$$

where $I(\cdot)$ denotes the indicator function, and $|\cdot|$ represents the cardinality of a set. In particular, we aim to enforce that FDR $\leq \alpha$ for a pre-specified level $\alpha \in (0, 1)$. The FDR represents the average proportion of selected units that do not meet alignment criteria, offering a direct measure of the risk incurred when deploying outputs in $\widehat{S}$. Apart from controlling the FDR, it is also desirable to select as many aligned units as possible, which corresponds to maximizing the power:

$$\text{Power} = \mathbb{E}\left[\frac{\sum_{j \in [m]} I\left\{A_{n+j} > c, j \in \widehat{S}\right\}}{\max\left(\sum_{j \in [m]} I\left\{A_{n+j} > c\right\}, 1\right)}\right].$$

### 3.2 CONFORMAL MIRROR STATISTICS FOR CONFORMAL ALIGNMENT

Following the conformalized selection framework (Jin & Candès, 2023a;b), we formalize alignment detection as a multiple testing problem. For each test unit $j \in [m]$, this induces the unit-level hypotheses

$$H_{0,j} : A_{n+j} \leq c, \qquad H_{1,j} : A_{n+j} > c.$$

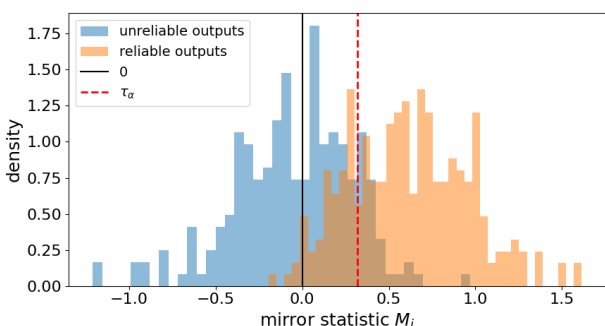

Figure 2: Empirical density of CMS. Outputs with mirror statistics above the cutoff $\tau_\alpha$ are selected.

We define the (true) null index set and alternative index set as

$$\mathcal{H}_0 := \{j \in [m] : A_{n+j} \leq c\}, \qquad \mathcal{H}_1 := \{j \in [m] : A_{n+j} > c\}.$$

Rejecting $H_{0,j}$ provides evidence that the alignment score of unit $j$ exceeds the threshold $c$. Thus aligned-unit selection reduces to simultaneously testing the hypotheses $\{H_{0,j}\}_{j=1}^m$.

Given the labeled data $\mathcal{D}_l$ with reference information, we begin by splitting $\mathcal{D}_l$ into two disjoint parts: a training set $\mathcal{D}_l^{\mathrm{tr}}$ and a calibration set $\mathcal{D}_l^{\mathrm{cal}}$. A prediction model $g : \mathcal{X} \to \mathbb{R}$ is then fitted on $\mathcal{D}_l^{\mathrm{tr}}$ to estimate the alignment score from the prompt $X$ and label $L$, which may also incorporate information from $f$. This yields predicted alignment scores $\widehat{A}_i = g(X_i)$ for $i \in [n]$ and $\widehat{A}_{n+j} = g(X_{n+j})$ for $j \in [m]$. The calibration pairs $\{(A_i, \widehat{A}_i)\}_{i \in \mathcal{D}_l^{\mathrm{cal}}}$ are subsequently used to construct the selection rule.

**Limitations of conventional methods.** Conformal $p$-values (Jin & Candès, 2023b; Gui et al., 2024) take the form

$$p_j = \frac{1 + \sum_{i \in \mathcal{D}_{\mathrm{cal}}} I\{A_i \leq c, \ \widehat{A}_i \geq \widehat{A}_{n+j}\}}{|\mathcal{D}_{\mathrm{cal}}| + 1},$$

and apply the BH procedure (Benjamini & Hochberg, 1995) to identify reliable sets:

$$\widehat{S} = \left\{ j \in [m] : p_j \leq \frac{\alpha k^*}{m} \right\}, \quad k^* = \max \left\{ k \in [m] : p_{(k)} \leq \frac{\alpha k}{m} \right\},$$

where $p_{(1)} \leq p_{(2)} \leq \cdots \leq p_{(m)}$ denote the $p$-values sorted in ascending order. From Gui et al. (2024), conformal $p$-value in this setting is valid only under the exchangeability assumption. This requirement is rarely satisfied in real applications where calibration and test units typically come from heterogeneous or non-exchangeable sources. Moreover, each $p_j$ takes one of the discrete levels $\{(r+1)/(n_{\mathrm{cal}}+1)\}_{r=0}^{n_{\mathrm{cal}}}$. When $m$ is large relative to $n_{\mathrm{cal}}$, these coarse $p$-values cannot distinguish fine differences among test scores, leading to substantial power loss in alignment problems with large test sets.

To overcome these limitations, we propose the Conformal Mirror Statistic, which incorporates both rank information and the magnitude differences between predicted scores. Specifically, for each $j \in [m]$ we define

$$M_j = \frac{\sum_{i \in \mathcal{D}_l^{\mathrm{cal}}} I\{A_i \leq c\} (\widehat{A}_{n+j} - \widehat{A}_i)}{\sum_{i \in \mathcal{D}_l^{\mathrm{cal}}} I\{A_i \leq c\}}. \tag{1}$$

Figure 2 shows the density distribution of conformal mirror statistics in simulated normal distribution data. The key intuition behind equation 1 is as follows. If a generated output for the test unit $X_{n+j}$ is unlikely to be aligned (i.e., $A_{n+j} \leq c$), then conditional on $A_i \leq c$, the distribution of $\widehat{A}_{n+j} - \widehat{A}_i$ should be symmetric around zero. Conversely, if $X_{n+j}$ is aligned (i.e., $A_{n+j} > c$), we expect $\widehat{A}_{n+j} - \widehat{A}_i$ to have a significantly positive magnitude given that $A_i \leq c$. This construction leads to two desirable properties of $M_j$:

(i) Under the null, the distribution of $M_j$ is symmetric about zero.

(ii) Larger values of $M_j$ indicate that the test unit $X_{n+j}$ is more likely to be aligned.

The symmetry guaranteed by Property (i) is particularly useful: it allows us to directly control the FDP without resorting to the BH procedure based on conformal $p$-values ordering. Consequently, our method does not suffer from the previously mentioned restriction that the test size $m$ must greatly exceed the calibration size $n$. We will rigorously establish these properties later.

Building on Property (i), for any threshold $\tau > 0$ the number of false positives under the null hypothesis can be controlled as

$$\#\{j \in \mathcal{H}_0 : M_j > \tau\} \approx \#\{j \in \mathcal{H}_0 : M_j < -\tau\} \lesssim \#\{j : M_j < -\tau\}, \qquad (2)$$

where $\#\{\cdot\}$ denotes the cardinality of a set. Consequently, this symmetry allows us to derive an approximate upper bound for the FDP directly:

$$\mathrm{FDP}(\tau) = \frac{\#\{j \in \mathcal{H}_0 : M_j > \tau\}}{\#\{j : M_j > \tau\} \vee 1} \lesssim \frac{\#\{j : M_j < -\tau\}}{\#\{j : M_j > \tau\} \vee 1}.$$

Together with Property (ii), this motivates using magnitude information rather than rank to select the discovery set $\widehat{S}$ at a target FDR level $\alpha \in (0, 1)$ as

$$\widehat{S} = \{j : M_j > \tau_\alpha\},$$

where the threshold $\tau_\alpha$ is determined by

$$\tau_\alpha = \min\left\{\tau > 0 : \widehat{\mathrm{FDP}}(\tau) := \frac{\#\{j : M_j < -\tau\}}{\#\{j : M_j > \tau\} \vee 1} \leq \alpha\right\}. \qquad (3)$$

The complete procedure is summarized in Algorithm 1. For Step 3, If the labeled data $\mathcal{D}_l$ are exchangeable, we adopt a random split.

---

**Algorithm 1** Conformal mirror statistics for testing alignment with FDR control

---

**Require:** Pre-trained foundation model $f$; alignment score function $\varphi$; labeled dataset $\mathcal{D}_l = \{X_i, L_i\}_{i=1}^n$; test dataset $\mathcal{D}_{\text{test}} = \{X_{n+j}\}_{j=1}^m$; method for fitting prediction model $g$; alignment level $c$; target FDR level $\alpha$.

**Ensure:** The final selected units set $\widehat{S}$.

1: Compute the alignment score $A_i = \varphi\left(f\left(X_i\right), L_i\right), \forall i \in \mathcal{D}_l$.
2: Partition $\mathcal{D}_l$ into two disjoint sets: the training set $\mathcal{D}_l^{\text{tr}}$ and the calibration set $\mathcal{D}_l^{\text{cal}}$.
3: Fit the alignment score predictor $g$ based on training dataset $\mathcal{D}_l^{\text{tr}}$.
4: Compute the predicted alignment score: $\widehat{A}_i \leftarrow g\left(X_i\right), \forall i \in \mathcal{D}_l^{\text{cal}}$, and $\widehat{A}_{n+j} \leftarrow g\left(X_{n+j}\right), \forall j \in \mathcal{D}_{\text{test}}$.
5: **for** $j \in [m]$ **do**
6:     Compute the conformal mirror statistics $M_j$ according to equation 1.
7: **end for**
8: For a given FDR level $\alpha \in (0, 1)$, determine the threshold $\tau_\alpha$ by equation 3.
9: Select the units set by $\widehat{S} = \{j : M_j > \tau_\alpha\}$.

---

Next, we provide formal justifications for Properties (i) and (ii) and rigorously establish FDR control. Property (ii) follows directly from the definition in equation 1, while Property (i) requires the following assumptions.

**Assumption 3.1** (Asymptotic Symmetry). *Fix a threshold $c \in \mathbb{R}$ with $\mathbb{P}(A_{n+j} \leq c) > 0$ for all $j \in [m]$. For each $X_j \in \mathcal{D}_l^{\text{cal}} \cup \mathcal{D}_{\text{test}}$, let $G_{0,j}$ denote the distribution of $g(X_j) - \mu_c$ conditional on $H_{0,j}$, where $\mu_c := \mathbb{E}[g(X_j) \mid H_{0,j}]$, and let $G_{0,j}^-$ be the reflected distribution defined by $G_{0,j}^-(B) = G_{0,j}(-B)$ for any Borel set $B$. We assume there exists a sequence $\eta_s = \eta_s(n_{\text{cal}}) \to 0$ as $n_{\text{cal}} \to \infty$, such that*

$$\sup_t |G_{0,j}(t) - G_{0,j}^-(t)| \leq \eta_s, \qquad \forall j \in [m].$$

**Assumption 3.2** (Uniform moment and tail regularity). *The calibration set $\{(X_i, L_i)\}_{i \in \mathcal{D}_l^{\mathrm{cal}}}$ and the test set $\{(X_{n+j}, L_{n+j})\}_{j \in [m]}$ are independent. The calibration estimated scores $\{\widehat{A}_i\}_{i \in \mathcal{D}_l^{\mathrm{cal}}}$ satisfy a uniform second-moment bound $\sup_{i \in \mathcal{D}_l^{\mathrm{cal}}} \mathbb{E}[\widehat{A}_i^2] < \infty$. For each test index $j$, define the tail function*

$$Q_j(\delta) := \mathbb{P}(\widehat{A}_{n+j} - \mu_c > \delta).$$

*We assume that each $Q_j$ is continuously differentiable in $\delta$ and that the derivatives are uniformly bounded:*

$$\sup_{j, \delta} |Q_j'(\delta)| < \infty.$$

*Remark* 3.3 (On the asymptotic symmetry assumption). Assumption 3.1 requires only that the null distribution $G_{0,j}$ of $g(X_j) - \mu_c$ be approximately symmetric in the sense that $\sup_t |G_{0,j}(t) - G_{0,j}^-(t)|$ vanishes with the calibration sample size. This is a mild condition. For example, if $g(X)$ is normally distributed then the symmetry holds exactly. More generally, for any distribution of $g(X)$ one can always construct a monotone transformation $F = S^{-1} \circ G$, where $G$ is the CDF of $g(X) \mid A \leq c$ and $S$ is any symmetric target CDF (e.g. the standard normal). Under this transformation, $F(g(X)) \mid A \leq c$ is exactly symmetric, and because $F$ is monotone our method can be applied after replacing $g$ by $F(g(X))$ without loss of generality.

*Remark* 3.4 (On distributional assumptions). The i.i.d. setting is only a special case of our Assumption 3.2. CMS requires neither identical distributions nor exchangeability, and our conditions are strictly weaker than those imposed in existing conformal $p$-value methods. We note that CMS can also remain valid under certain forms of weak dependence (e.g., $\alpha$-mixing sequences), although for generality and clarity we impose independence in our theoretical development.

We are now ready to establish the asymptotic symmetry of the conformal mirror statistic.

**Proposition 3.5** (Asymptotic symmetry of $M_j$). *Under Assumptions 3.1 and 3.2, define $n_{\mathrm{cal}} := |\mathcal{D}_l^{\mathrm{cal}}|$. Under $H_{0,j}$, the mirror statistic $M_j$ is asymptotically symmetric about 0; that is,*

$$\lim_{n_{\mathrm{cal}} \to \infty} \mathbb{P}(M_j \leq t \mid H_{0,j}) = 1 - \lim_{n_{\mathrm{cal}} \to \infty} \mathbb{P}(M_j \leq -t \mid H_{0,j}), \qquad \forall t \in \mathbb{R}.$$

With Proposition 3.5, we can prove that Algorithm 1 achieves FDR control:

**Theorem 3.6.** *Assume that the calibration size $n_{cal}$ grows polynomially with $m_0 := |\mathcal{H}_0|$, i.e. $n_{\mathrm{cal}} \asymp m_0^{\beta}$ for some $\beta > 0$. For any nominal FDR level $\alpha \in (0, 1)$, assume that there exists a constant $\tau > 0$ such that $\mathbb{P}(\mathrm{FDP}(\tau) \leq \alpha) \to 1$ as $m \to \infty$. Then, under Assumptions 3.1 and 3.2, Algorithm 1 satisfies*

$$\mathrm{FDP}(\tau_\alpha) \leq \alpha + o_m(1) \quad and \quad \limsup_{n_{cal}, \, m \to \infty} \mathrm{FDR}(\tau_\alpha) \leq \alpha.$$

*Remark* 3.7 (On Asymptotic Guarantees). The result above establishes asymptotic FDR control. Compared with the finite-sample guarantee of classical conformal methods, this reflects a tradeoff: CMS can work with the much weaker non-exchangeable condition (see Remark 3.4), which allows heteroskedasticity in the data. For this reason, we view the absence of a finite-sample theorem as a modest weakness. Moreover, both our simulation studies and real-data analysis show that CMS achieves reliable finite-sample FDR control in practice, suggesting that the asymptotic guarantee captures the method's practical behavior.

We also provide the following proposition to show that our method achieves the asymptotic power.

**Proposition 3.8** (Asymptotic Power of CMS). *Under Assumptions 3.1 and assume that the calibration set $\{(X_i, L_i)\}_{i \in \mathcal{D}_l^{\mathrm{cal}}}$ and the test set $\{(X_{n+j}, L_{n+j})\}_{j \in [m]}$ are independent and identically distributed. Under the i.i.d. population model where $H_0$ denotes the null population ($A \leq c$) and $H_1$ the alternative population ($A > c$), let $M = g(X) - \mu_c$ with $\mu_c = \mathbb{E}[g(X) \mid H_0]$, and assume $M$ is continuously distributed.*

*At the population level, the false discovery rate associated with a threshold $\tau$ is defined as*

$$\mathrm{FDR}(\tau) = \frac{\mathbb{P}(M > \tau, A \leq c)}{\mathbb{P}(M > \tau)}.$$

*Further assume that* $\mathrm{FDR}(\tau)$ *is strictly decreasing at the oracle threshold* $\tau^*(\alpha) = \inf\{\tau : \mathrm{FDR}(\tau) \leq \alpha\}$ *(see Appendix D), so that the CMS threshold* $\tau_\alpha$ *converges in probability to* $\tau^*(\alpha)$ *as* $n_{\mathrm{cal}}, m \to \infty$. *Consequently,*

$$\lim_{n_{\mathrm{cal}}, m \to \infty} \mathrm{Power}(\tau_\alpha) = \mathbb{P}\big(M > \tau^*(\alpha) \mid H_1\big), \tag{4}$$

$$\lim_{n_{\mathrm{cal}}, m \to \infty} \frac{1}{m} \sum_{j=1}^{m} I\{M_j > \tau_\alpha, \, A_{n+j} > c\} = \mathbb{P}\big(M > \tau^*(\alpha), \, A > c\big). \tag{5}$$

*Remark* 3.9 (On the Role of $g$ and Power Equivalence). A more informative $g$ assigns larger values under $H_1$ and smaller values under $H_0$, which makes $\mathbb{P}(M > \tau, \, A \leq c)$ small. Consequently, the oracle threshold $\tau^*(\alpha)$ required to satisfy the FDR constraint also becomes small. Combined with a larger $M$ under $H_1$, the power $\mathbb{P}(M > \tau^*(\alpha) \mid H_1)$ becomes larger. Moreover, the power expression has the same structural form as that of conformal $p$-value methods, since both approaches result in comparing a single data-dependent score against a threshold. It means that CMS remains valid under non-exchangeable and heterogeneous settings, while in the i.i.d. case its power does not suffer compared to conformal $p$-value methods. A detailed comparison is provided in Appendix E.

*Remark* 3.10 (On the Role of $g$ and Power Equivalence).

**Generality of the framework.** Theorem 3.6 establishes that our method achieves asymptotic FDR control under mild assumptions. Importantly, these guarantees depend only on generic properties of the alignment predictor $g$, rather than on the specific form of the underlying model $f$. This model-agnostic design enables the framework to generalize across different architectures and application domains. Moreover, because the procedure is distribution-free and requires only a small labeled dataset, it remains statistically valid even in heterogeneous real-world scenarios. In addition, the estimation accuracy of the alignment scores does not affect FDR validity, as the guarantees rely only on the symmetry property of $g$ rather than its predictive precision. Because a single data split can introduce variance, we also construct a more robust multi-splitting version of CMS. The method and supporting results are provided in Appendix F.

## 4 SIMULATIONS

In this section, we will demonstrate the performance of CMS compared with baseline method under homoscedastic and heteroscedastic settings.

We generate covariates $X \in \mathbb{R}^{20}$ independently as $X_j \sim \mathrm{Unif}[-1, 1]$. A latent continuous outcome is defined by $A = \mu(X) + \varepsilon$, $\varepsilon \sim N(0, \sigma(X)^2)$, where the nonlinear signal is

$$\mu(X) = 2\big(X_1 X_2 + X_3^2 + e^{X_4 - 1} - 1\big).$$

Heterogeneity is introduced through the noise level $\sigma(X)$. We consider two simulation settings:

(i) a **homoscedastic** case $\sigma(X) = 1.5$,

(ii) a **heteroscedastic** case $\sigma(X) = \frac{5.5 - |\mu(X)|}{2}$. Although the covariates X are generated i.i.d., the heteroscedastic noise structure makes the alignment scores non-exchangeable, which violates the exchangeability assumption required for conformal p-value methods.

Throughout all experiments we set the true alignment threshold to $c = 0.2$. We generate 2000 samples and assign 80% to the test set, which is a largely unlabeled scenario. To construct the alignment predictor $g(X)$, we fit three standard models on $(X, A)$: Gradient Boosting Regression (GBR), Random Forest Regression (RF), and Support Vector Regression (SVR).

Figures 3 and 4 compare FDR and power under the homoscedastic setting (i) and the heterogeneous setting (ii). Across all base learners, CMS consistently achieves more stable FDR control and higher power in both settings. This reflects a key practical benefit of mirror statistics: unlike conformal $p$-values, CMS does not rely on exchangeability assumptions, and therefore remains reliable even when noise levels and signal structures vary across the covariate space. In contrast, conformal $p$-values exhibit clear FDR inflation and reduced power, especially in Setting (ii), where input-dependent variance breaks the calibration–test comparability required by the ranking-based construction. Overall, CMS is substantially more robust in nonlinear and heterogeneous environments, precisely where the conformal $p$-value method tends to degrade.

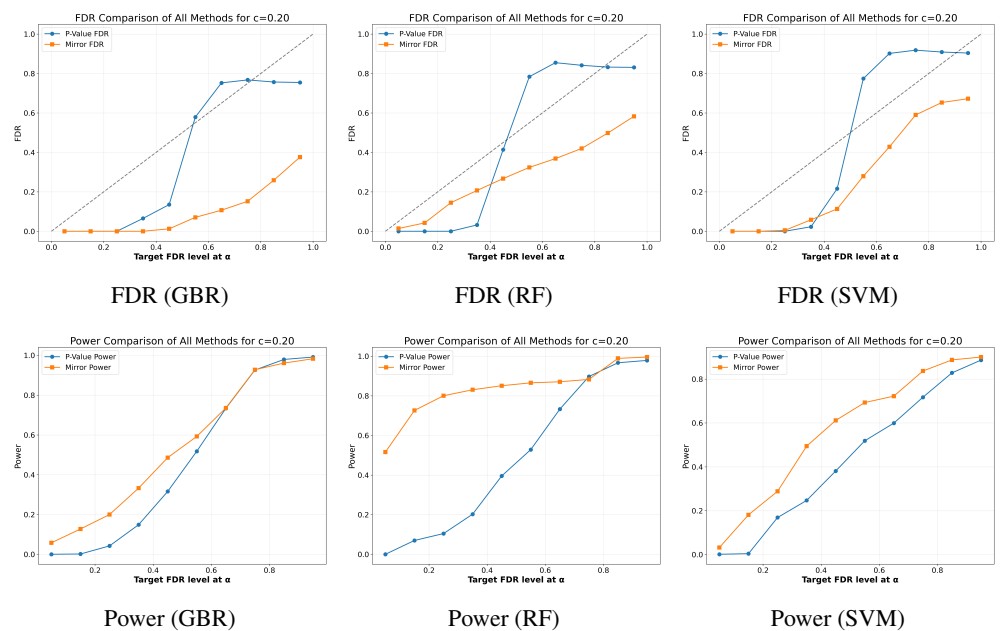

Figure 3: FDR (top) and power (bottom) using GBR, RF, and SVM under Setting (i) $\sigma(X) = 1.5$ at $c = 0.20$.

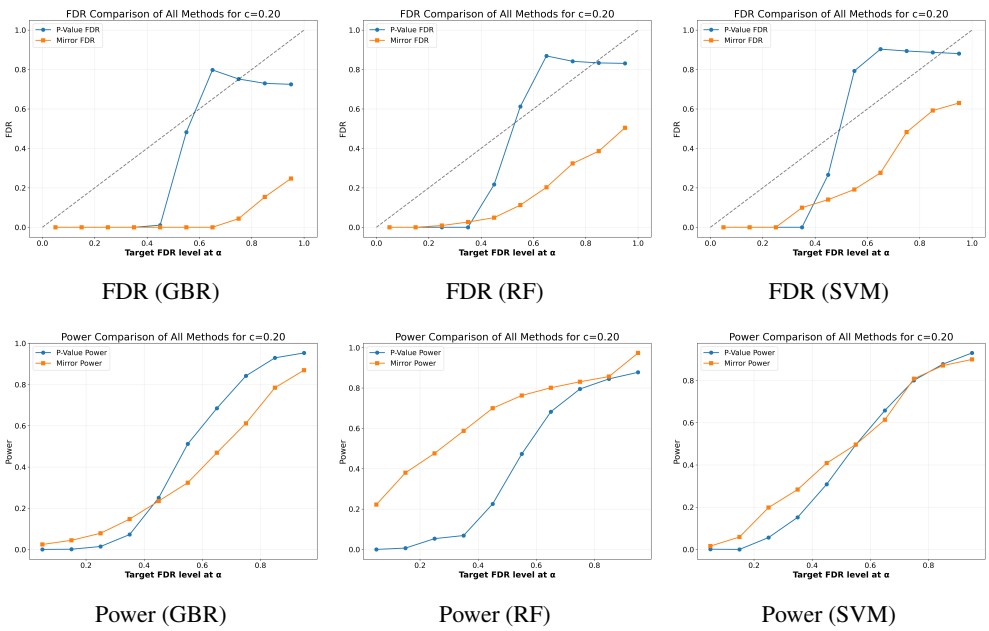

Figure 4: FDR (top) and power (bottom) using GBR, RF, and SVM under Setting (ii) $\sigma(X) = \frac{5.5 - |\mu(X)|}{2}$ at $c = 0.20$.

## 5 REAL DATA EXPERIMENTS

### 5.1 EXPERIMENTAL SETUP

In this section, we will focus on the application on high-stakes clinical decision-making such as dynamic treatment regimes (DTRs). We evaluate the method using a cohort of sepsis patients derived from the Medical Information Mart for Intensive Care (MIMIC)-III dataset, focusing on early-stage

treatment planning within a 72-hour clinical window surrounding sepsis onset (Singer et al., 2016). For each patient we input the multimodel EHR data including the tabular data (e.g., vitals, labs, demographics) and clinical notes into the model. We use the multimodal recommendation model based on a Transformer architecture (Shen et al., 2025) to generate a treatment $f(X_i)$ among the treatment space, which consists of 25 discrete treatment combinations $\{T_i\}_{i=1}^{25}$ formed by intravenous fluid and vasopressor dosages.

It is important to note that, unlike the idealized exchangeable assumption, real-world EHR data are not independent across patients. Patients treated in the same ICU may share correlated treatment policies, hospital practices, or clinical environments, leading to heteroskedasticity or weak dependence in their observed trajectories.

**Alignment Score** $\varphi(X_i)$**.** We follow Shen et al. (2025) and define the alignment score as the divergence between the base model's predictive distribution and that of a student network fine-tuned only on survivor trajectories. Survivor trajectories are used not as unbiased ground truth but as a lower-noise reference: their treatment patterns tend to be more stable and less confounded by terminal deterioration than those from deceased patients. This design does not discard or ignore deceased patients. Instead, the divergence between the two models is explicitly interpreted as an uncertainty signal, reflecting label ambiguity in noisier mortality trajectories. In this way, the survivor-based refinement serves as a proxy for identifying unstable supervision rather than introducing selection bias, and the subsequent FDR-controlled selection steps ensure that such uncertainty is properly incorporated.

## 5.2 RESULTS ANALYSIS

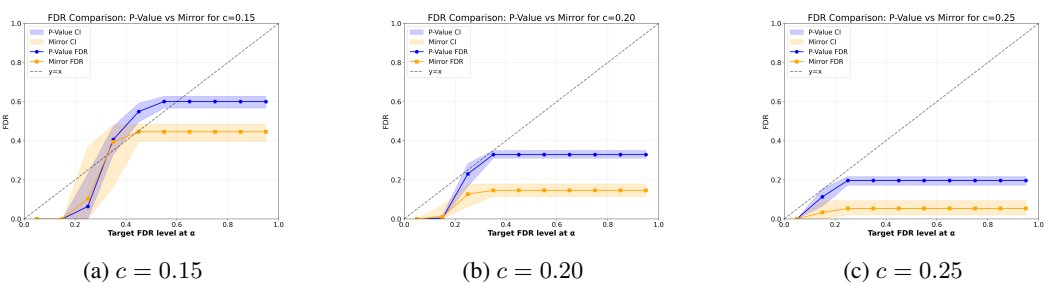

(a) $c = 0.15$      (b) $c = 0.20$      (c) $c = 0.25$

Figure 5: FDR comparison across different $c$ values.

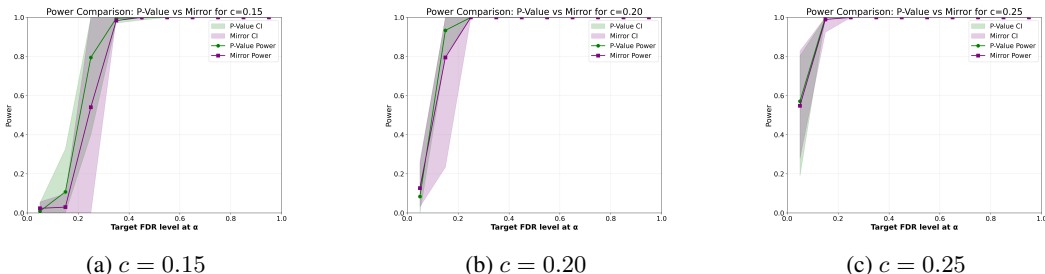

(a) $c = 0.15$      (b) $c = 0.20$      (c) $c = 0.25$

Figure 6: Power comparison across different $c$ values.

Across 100 independent experiments and a range of uncertainty thresholds $c$, we compare CMS with the standard conformal $p$-value method in terms of both FDR and power. Figure 5 shows the realized FDR across three representative thresholds $c \in \{0.15, 0.20, 0.25\}$. CMS exhibits stable and accurate FDR control across all target levels $\alpha$, with empirical FDR curves closely tracking the nominal line. In contrast, conformal $p$-values tend to yield higher FDR, reflecting their sensitivity to heteroscedasticity and the violation of exchangeability.

Figure 6 presents the corresponding power results. The two methods achieve broadly comparable power across all examined $c$ values, with only minor differences attributable to variance in the align-

ment score or threshold structure. This confirms that CMS does not sacrifice detection ability while providing more reliable FDR control.

Together, these findings demonstrate that CMS provides accurate finite-sample FDR control and competitive power across a wide range of uncertainty thresholds. This stability persists even in heterogeneous settings where conformal $p$-values can become miscalibrated, highlighting the robustness of CMS in practical, non-exchangeable data environments.

# 6 CONCLUSION

In this work, we introduced Conformal Mirror Statistics (CMS), a flexible and distribution-free framework for model alignment with rigorous FDR control. By leveraging mirror statistics, CMS eliminates the need for large calibration sets and remains valid under far weaker conditions than conventional conformal $p$-value methods. In particular, our theory accommodates non-exchangeable and heteroskedastic data settings where exchangeability-based conformal $p$-values typically fail. Empirical studies on simulations and sepsis treatment recommendation further demonstrate that CMS achieves competitive efficiency while maintaining strict FDR guarantees.

Looking forward, a promising direction is to integrate CMS with modern large-scale machine learning pipelines, for example in reinforcement learning or large language models. Also we could extend CMS to online or streaming environments, where both calibration and test distributions evolve over time. Such extensions would broaden CMS's applicability in dynamically changing, real-world decision-making systems.

## ETHICS STATEMENT

This study uses the de-identified MIMIC-III database, which is accessible to credentialed researchers who have completed the required data use agreement and human-subjects training. No personally identifiable information is included. All experiments were conducted in compliance with institutional and ethical guidelines. Our methods are intended for uncertainty quantification and model alignment in predictive modeling. Deployment in clinical practice should be approached cautiously to avoid misuse, such as replacing clinical judgment or reinforcing existing biases.

## REPRODUCIBILITY STATEMENT

We provide detailed descriptions of our methodology, assumptions, and hyperparameter settings in the main text and appendix. Pseudocode and theoretical proofs are included to ensure transparency. The source code is currently under preparation for public release, and we aim to make it available in the future. The MIMIC-III data cannot be shared directly due to access restrictions, but researchers with appropriate credentials and data access should be able to reproduce our experiments by following the provided procedures.

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

## A USE OF LARGE LANGUAGE MODELS (LLMS)

During the preparation of this manuscript, we used an LLM-based writing assistant (OpenAI's Chat-GPT) for language polishing. Specifically, the tool was employed to improve grammar, rephrase sentences for clarity, and refine the overall readability of the text. All ideas, technical content, theoretical results, and experimental analyses are solely the work of the authors, who take full responsibility for the correctness and integrity of the paper.

## B PROOF OF PROPOSITION 3.5

*Proof.* Write $\widehat{\mu}_c := \frac{\sum_{i \in \mathcal{D}_l^{\mathrm{cal}}} I\{A_i \leq c\} \widehat{A}_i}{\sum_{i \in \mathcal{D}_l^{\mathrm{cal}}} I\{A_i \leq c\}}$, we have

$$
\begin{aligned}
M_j &= \widehat{A}_{n+j} - \frac{\sum_{i \in \mathcal{D}_l^{\mathrm{cal}}} I\{A_i \leq c\} \widehat{A}_i}{\sum_{i \in \mathcal{D}_l^{\mathrm{cal}}} I\{A_i \leq c\}} \\
&= (\widehat{A}_{n+j} - \mu_c) - (\widehat{\mu}_c - \mu_c) \\
&= \widehat{A}_{n+j} - \mu_c + O_p(n_{\mathrm{cal}}^{-1/2}),
\end{aligned}
$$

where the last equality comes from Assumption 3.2 and Chebyshev inequality.

Assumption 3.2 implies that the unconditional density $Q'_j$ is uniformly bounded. Since $H_{0,j} : A_{n+j} \leq c$ satisfies $\mathbb{P}(A_{n+j} \leq c) > 0$, conditioning on $H_{0,j}$ only rescales the unconditional density by the factor $1/\mathbb{P}(A_{n+j} \leq c)$, the conditional density under $H_{0,j}$ is also uniformly bounded. Hence each $G_{0,j}$ is Lipschitz with a universal constant $L$.

Thus for any $t$,

$$
P(M_j \leq t) = G_{0,j}(t) + O_p(n_{\mathrm{cal}}^{-1/2}), \qquad P(M_j \leq -t) = 1 - G_{0,j}^-(t) + O_p(n_{\mathrm{cal}}^{-1/2}).
$$

From Assumption 3.1, we have

$$
\sup_t |P(M_j \leq t) - (1 - P(M_j \leq -t))| \eta_s + O_p(n_{\mathrm{cal}}^{-1/2}),
$$

where $\eta_s \to 0$ when $n_{\mathrm{cal}} \to \infty$. This proves the approximate symmetry of $M_j$. $\square$

## C PROOF OF THEOREM 3.6

We begin by proving an essential proposition, which serves as a key step toward Theorem 3.6.

**Proposition C.1.** *(Weak dependence) Under Assumptions 3.1 and 3.2, there exist constants $\omega > 0$ and $\gamma \in (0, 2)$ such that*

$$
\mathrm{Var}\left( \sum_{j \in \mathcal{H}_0} I(M_j > \tau) \right) \leq \omega m_0^\gamma, \ \forall \tau \in \mathbb{R}.
$$

*Proof of Proposition C.1.* Let $S(\tau) := \sum_{j \in \mathcal{H}_0} I\{M_j > \tau\}$. By the law of total variance,

$$
\mathrm{Var}\big(S(\tau)\big) = \mathbb{E}\big[\mathrm{Var}(S(\tau) \mid \mathcal{D}_l^{\mathrm{cal}})\big] + \mathrm{Var}\big(\mathbb{E}[S(\tau) \mid \mathcal{D}_l^{\mathrm{cal}}]\big).
$$

Given $\mathcal{D}_l^{\mathrm{cal}}$, $\Delta := \widehat{\mu}_c - \mu_c$ is a constant and $M_j = (\widehat{A}_{n+j} - \mu_c) - \Delta$ depends only on the $j$-th test sample. By Assumption 3.2, the test samples are independent, hence $\{I(M_j > \tau)\}_{j \in \mathcal{H}_0}$ are conditionally independent Bernoulli with success probability $p_{j,\Delta}(\tau) := \mathbb{P}(\widehat{A}_{n+j} - \mu_c > \tau + \Delta \mid \mathcal{D}_l^{\mathrm{cal}})$. Therefore,

$$
\mathrm{Var}(S(\tau) \mid \mathcal{D}_l^{\mathrm{cal}}) = \sum_{j \in \mathcal{H}_0} p_{j,\Delta}(\tau)\big(1 - p_{j,\Delta}(\tau)\big) \leq \frac{m_0}{4},
$$

and thus $\mathbb{E}[\mathrm{Var}(S(\tau) \mid \mathcal{D}_l^{\mathrm{cal}})] \leq \frac{m_0}{4}$.

Remember $Q_j(\delta) = \mathbb{P}(\widehat{A}_j - \mu_c > \delta)$, so that $p_{j,\Delta}(\tau) = Q_j(\tau + \Delta)$. By Assumption 3.2, $Q_j$ is differentiable with bounded derivative $Q_j'$. A first-order expansion yields

$$p_{j,\Delta}(\tau) = Q_j(\tau) + Q_j'(\tau)\Delta + r_j(\Delta), \qquad |r_j(\Delta)| \leq C \Delta^2,$$

for some constant $C$. Hence

$$\mathbb{E}[S(\tau) \mid \mathcal{D}_l^{\mathrm{cal}}] = \sum_{j \in \mathcal{H}_0} p_{j,\Delta}(\tau),$$

and therefore

$$\mathrm{Var}\big(\mathbb{E}[S(\tau) \mid \mathcal{D}_l^{\mathrm{cal}}]\big) = \mathrm{Var}\Big( \sum_{j \in \mathcal{H}_0} p_{j,\Delta}(\tau)\Big).$$

Since each $p_{j,\Delta}(\tau) = Q_j(\tau + \Delta)$ is uniformly Lipschitz in $\Delta$,

$$\mathrm{Var}\Big( \sum_{j \in \mathcal{H}_0} p_{j,\Delta}(\tau)\Big) \lesssim m_0^2 \, \mathrm{Var}(\Delta).$$

Since $\Delta = \widehat{\mu}_c - \mu_c$ is the sample mean of $Y_i := \widehat{A}_i \, I(A_i \leq c)$ over the $n_{\mathrm{cal},c} := \sum_{i \in \mathcal{D}_{\mathrm{cal}}} I(A_i \leq c)$ calibration points with $A_i \leq c$, we have under Assumption 3.2, the calibration estimated scores $\{\widehat{A}_i\}$ are independent with a uniform second-moment bound $\sup_i \mathbb{E}[\widehat{A}_i^2] < \infty$. Therefore,

$$\mathrm{Var}(\widehat{\mu}_c) = \frac{1}{n_{\mathrm{cal},c}^2} \sum_{i:A_i \leq c} \mathrm{Var}(Y_i) = O\Big( \frac{1}{n_{\mathrm{cal},c}}\Big),$$

which gives $\mathrm{Var}(\Delta) = O(1/n_{\mathrm{cal},c})$.

By the law of large numbers, $n_{\mathrm{cal},c} \asymp n_{\mathrm{cal}} \cdot \mathbb{P}(A \leq c)$, so that

$$\mathrm{Var}\big(\mathbb{E}[S(\tau) \mid \mathcal{D}_l^{\mathrm{cal}}]\big) = O\Big( \frac{m_0^2}{n_{\mathrm{cal}}}\Big).$$

Combining with the conditional variance bound, we obtain

$$\mathrm{Var}(S(\tau)) \;\leq\; C_1 \, m_0 \;+\; C_2 \, \frac{m_0^2}{n_{\mathrm{cal}}}.$$

Since we have the calibration size grows polynomially with $m_0$:

$$n_{\mathrm{cal}} \asymp m_0^\beta, \qquad \beta > 0.$$

Then the second term satisfies

$$\frac{m_0^2}{n_{\mathrm{cal}}} = O(m_0^{2-\beta}),$$

so that

$$\mathrm{Var}(S(\tau)) \;\lesssim\; m_0 + m_0^{2-\beta}.$$

In particular, when $\beta = 1$ (e.g., a fixed split ratio), we obtain $\mathrm{Var}(S(\tau)) \lesssim m_0$, corresponding to $\gamma = 1$. More generally, for any $\beta > 0$, the bound holds with $\gamma = \max\{1, \, 2 - \beta\} \in (0, 2)$, i.e.

$$\mathrm{Var}\left( \sum_{j \in \mathcal{H}_0} I\{M_j > \tau\} \right) \leq \omega \, m_0^\gamma, \qquad \forall \tau \in \mathbb{R}.$$

$\square$

With Propositions 3.5 and C.1 in place, Proposition 2.2 of Dai et al. (2023b) directly yields Theorem 3.6.

# D    PROOF OF PROPOSITION 3.8

*Proof.* By Theorem 3.6, under Assumption 3.1 and i.i.d. the CMS estimator $\widehat{\mathrm{FDP}}(\tau)$ is a consistent estimator of $\mathrm{FDR}(\tau)$ at every fixed $\tau > 0$. Hence, for any fixed threshold $\tau$,

$$\widehat{\mathrm{FDP}}(\tau) \xrightarrow{P} \mathrm{FDR}(\tau) \qquad (n_{\mathrm{cal}}, m \to \infty).$$

Assumption 3.2 ensures that the mirror statistic $M$ has a continuous distribution, so its empirical distribution satisfies Glivenko–Cantelli uniform convergence. Consequently,

$$\sup_{\tau \in [0,T]} \left| \widehat{\mathrm{FDP}}(\tau) - \mathrm{FDR}(\tau) \right| \xrightarrow{P} 0, \qquad \text{for every fixed } T > 0.$$

Define

$$\psi(\tau) = \mathrm{FDR}(\tau) - \alpha, \qquad \widehat{\psi}(\tau) = \widehat{\mathrm{FDP}}(\tau) - \alpha.$$

By the uniform convergence above, $\widehat{\psi} \to \psi$ uniformly on compact intervals. The mild technical condition that $\mathrm{FDR}(\tau)$ is strictly decreasing at

$$\tau^*(\alpha) = \inf\{\tau : \mathrm{FDR}(\tau) \le \alpha\}$$

ensures that $\tau^*(\alpha)$ is the unique first zero-crossing of $\psi$. Classical stability results for zero-crossings under uniform convergence now yield

$$\tau_\alpha = \inf\{\tau : \widehat{\mathrm{FDP}}(\tau) \le \alpha\} \xrightarrow{P} \tau^*(\alpha).$$

Next consider the empirical proportion of true aligned selections under the random threshold $\tau_\alpha$. Since $\tau_\alpha \to \tau^*(\alpha)$ and $M$ has a continuous distribution, the probability that $M$ lies in any shrinking neighborhood of $\tau^*(\alpha)$ vanishes. Thus the difference

$$I\{M_j > \tau_\alpha, \ A_{n+j} > c\} - I\{M_j > \tau^*(\alpha), \ A_{n+j} > c\}$$

contributes negligibly in the limit. Because $M_j = g(X_{n+j}) - \widehat{\mu}_c$ and the empirical baseline $\widehat{\mu}_c$ satisfies $\widehat{\mu}_c \xrightarrow{p} \mu_c := \mathbb{E}[g(X) \mid A \le c]$, we obtain the convergence

$$M_j \xrightarrow{p} M := g(X) - \mu_c.$$

Consequently, the indicators $I\{M_j > \tau_\alpha, \ A_{n+j} > c\}$ converge in distribution to $I\{M > \tau^*(\alpha), \ A > c\}$. By the law of large numbers,

$$\frac{1}{m} \sum_{j=1}^{m} I\{M_j > \tau_\alpha, \ A_{n+j} > c\} \xrightarrow{P} \mathbb{P}(M > \tau^*(\alpha), \ A > c).$$

Finally, the CMS power satisfies

$$\mathrm{Power}(\tau_\alpha) = \mathbb{E}\left[ \frac{m^{-1} \sum_{j=1}^{m} I\{M_j > \tau_\alpha, \ A_{n+j} > c\}}{m^{-1} \sum_{j=1}^{m} I\{A_{n+j} > c\}} \right].$$

The denominator converges to $\mathbb{P}(A > c) > 0$, and the numerator converges as above. Slutsky's theorem yields

$$\lim_{n_{\mathrm{cal}}, m \to \infty} \mathrm{Power}(\tau_\alpha) = \mathbb{P}(M > \tau^*(\alpha) \mid H_1),$$

together with

$$\lim_{n_{\mathrm{cal}}, m \to \infty} \frac{1}{m} \sum_{j=1}^{m} I\{M_j > \tau_\alpha, \ A_{n+j} > c\} = \mathbb{P}(M > \tau^*(\alpha), \ A > c).$$

Since no valid level-$\alpha$ procedure can exceed the oracle power $\mathbb{P}(M > \tau^*(\alpha) \mid H_1)$, CMS achieves the maximal asymptotic power.  □

# E STRUCTURAL EQUIVALENCE BETWEEN CONFORMAL ALIGNMENT AND CMS

First, we will rewrite the power of conformal p-value.

**Proposition E.1.** *Consider the conformal p-value*

$$p_j = \frac{1 + \sum_{i \in D_{\text{cal}}} I\{A_i \leq c,\ g(X_i) \geq g(X_{n+j})\}}{|D_{\text{cal}}| + 1}.$$

*The asymptotic power of conformal p-value admits the expression*

$$\lim_{n,m \to \infty} \text{Power} = \frac{\mathbb{P}\{g(X) \geq T^*,\ H_1\}}{\mathbb{P}(A > c)} \qquad \text{for some constant } T^*.$$

*Proof.* Let $V(x, a) = 2\overline{M} \cdot I\{a > c\} - g(x)$ for some constant $\overline{M} > \sup_x |g(x)|$. We begin by rewriting the $p$-value as

$$p_j = \frac{1 + \sum_{i \in D_{\text{cal}}} I\{V_i \leq \widehat{V}_{n+j}\}}{|D_{\text{cal}}| + 1},$$

so that $p_j$ is a monotone nondecreasing function of $\widehat{V}_{n+j}$.

apply the Proposition 2.10 in Jin & Candès (2023b),we can get the asymptotic power taking the form

$$\frac{\mathbb{P}\{g(X) \geq T^*,\ A > c\}}{\mathbb{P}(A > c)},$$

for some constant $T^*$, which completes the proof. $\qquad\square$

Because the CMS statistic satisfies

$$M(X) = g(X) - \frac{\sum_{i \in \mathcal{D}_l^{\text{cal}}} I\{A_i \leq c\}\, \widehat{A}_i}{\sum_{i \in \mathcal{D}_l^{\text{cal}}} I\{A_i \leq c\}},$$

the rejection event $M(X) > \tau^*(\alpha)$ is equivalent to

$$g(X) > \tau^*(\alpha) + \frac{\sum_{i \in \mathcal{D}_l^{\text{cal}}} I\{A_i \leq c\}\, \widehat{A}_i}{\sum_{i \in \mathcal{D}_l^{\text{cal}}} I\{A_i \leq c\}}.$$

Hence the asymptotic CMS power can be written as

$$\lim_{n_{\text{cal}}, m \to \infty} \text{Power}_{\text{CMS}} = \mathbb{P}\left( g(X) > \tau^*(\alpha) + \frac{\sum_{i \in \mathcal{D}_l^{\text{cal}}} I\{A_i \leq c\}\, \widehat{A}_i}{\sum_{i \in \mathcal{D}_l^{\text{cal}}} I\{A_i \leq c\}} \,\middle|\, H_1 \right).$$

In summary, both conformal $p$-values and CMS admit the same structural form for their asymptotic power. Conformal $p$-values yield a selection rule of the form

$$g(X) \geq \tau^*_{\text{cp}}, \qquad \Longrightarrow \qquad \lim_{n,m \to \infty} \text{Power}_{\text{cp}} = \mathbb{P}\big(g(X) \geq \tau^*_{\text{cp}} \mid H_1\big),$$

while CMS induces a shifted threshold,

$$g(X) > \tau^*_{\text{cms}} + C_{\text{cal}}, \qquad \Longrightarrow \qquad \lim_{n_{\text{cal}}, m \to \infty} \text{Power}_{\text{cms}} = \mathbb{P}\big(g(X) > \tau^*_{\text{cms}} + C_{\text{cal}} \mid H_1\big).$$

Thus both procedures share the same functional form

$$\mathbb{P}\{\, g(X) > \text{threshold} \mid H_1 \,\},$$

differing only in the data–dependent threshold. Since the CMS threshold explicitly incorporates the alignment boundary $c$ through the mirror construction, it is generally more adaptive to the underlying distributional structure than the conformal $p$-value threshold.

## F   ROBUST CONFORMAL ALIGNMENT

Although Algorithm 1 controls the FDR at the desired level $\alpha$, two important concerns remain. First, splitting the data inflates the variance of the predicted alignment scores compared to competing methods that fully utilize the data. Second, the conformal alignment selections produced by Algorithm 1 may be unstable and vary substantially across different random splits.

To address these issues, we propose a multiple data-splitting procedure that aggregates selection results from independent replications of Algorithm 1. Specifically, we run the algorithm $K$ times with random sample splits, obtaining conformal mirror statistics $\{M_j^{(k)}\}_{j\in[m]}$, thresholds $\tau_\alpha^{(k)}$ and the selected set $\widehat{S}^{(k)}$. We then aggregate these results using the framework of $e$-values (Vovk & Wang, 2021), a recently proposed alternative to $p$-values that allows valid inference under arbitrary dependence. A nonnegative random variable $e$ is called an $e$-value if $\mathbb{E}[e] \leq 1$ under the null, with larger values indicating stronger evidence against it. For instance, we should reject when $e \geq 1/\alpha$ controls type-I error at level $\alpha$.

For each test unit $X_{n+j}$, we construct its aggregated $e$-value as the average of the $K$ replications:

$$
e_j = \frac{1}{K}\sum_{k=1}^{K} e_j^{(k)}, \qquad e_j^{(k)} = \text{weight}_j^{(k)} \cdot I\{j \in \widehat{S}^{(k)}\}. \tag{6}
$$

Inspired by Ren & Barber (2024) and Dai et al. (2023b), we construct two $e$-value forms tailored to our setting: the derandomized version, $e_j^{(k)} = \frac{m\cdot I(M_j^{(k)}>\tau_\alpha^{(k)})}{1+\sum_{j\in[m]} I(M_j^{(k)}\leq-\tau_\alpha^{(k)})}$, and the inclusion-rate, $e_j^{(k)} = \frac{m\cdot I(M_j^{(k)}>\tau_\alpha^{(k)})}{\alpha\cdot\sum_{j\in[m]} I\{M_j^{(k)}\geq\tau_\alpha^{(k)}\}}$, which differ in how they normalize by the number of selected units.

Given the aggregated $e_j$ values, we apply the e-BH procedure (Algorithm 2) to control FDR. By Wang & Ramdas (2022), e-BH ensures FDR $\leq \alpha \cdot m_0/m \leq \alpha$, as summarized in the following theorem.

---

**Algorithm 2** e-BH Procedure for FDR Control

---

**Require:** $e$-values: $e_1, e_2, \ldots, e_m$, FDR level: $\alpha$.
**Ensure:** Final selected set $\widehat{S}$.
  1: Sort the $e$-values in decreasing order: $e_{(1)} \geq e_{(2)} \geq \cdots \geq e_{(m)}$,
  2: Set $d = \max\left\{j : e_{(j)} \geq \frac{m}{\alpha j}\right\}$,
  3: Return $\widehat{S} = \{j : e_j \geq \frac{m}{\alpha d}\}$.

---

**Theorem F.1** (e-BH FDR control). *Suppose the values $e_1, e_2, \ldots, e_m$ we construct for aggregation satisfy the $e$-value criterion $\mathbb{E}(e_j) \leq 1$ for all $j \in \mathcal{H}_0$ (or the relaxed $e$-value criterion $\sum_{j\in\mathcal{H}_0}\mathbb{E}(e_j) \leq m$). Then the e-BH procedure in Algorithm 2 guarantees FDR $\leq \alpha$ for any level $\alpha \in (0,1)$.*

We demonstrate the simulation results for multiple-splitting mirror statistics here, with the same setting from Section 4.

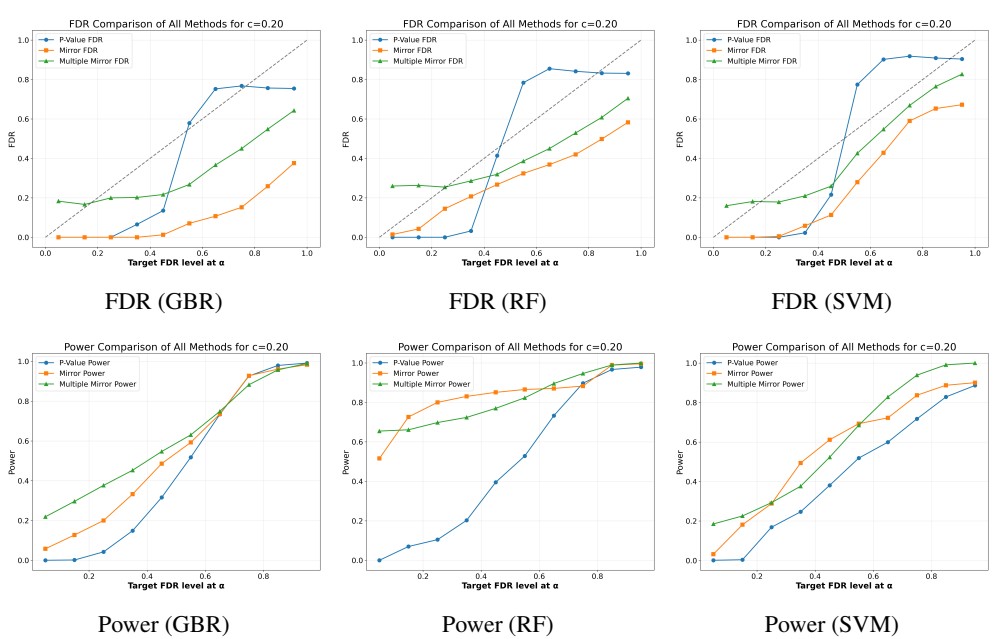

Figure 7: FDR (top) and power (bottom) using GBR, RF, and SVM under Setting (i) $\sigma(X) = 1.5$ at $c = 0.20$.

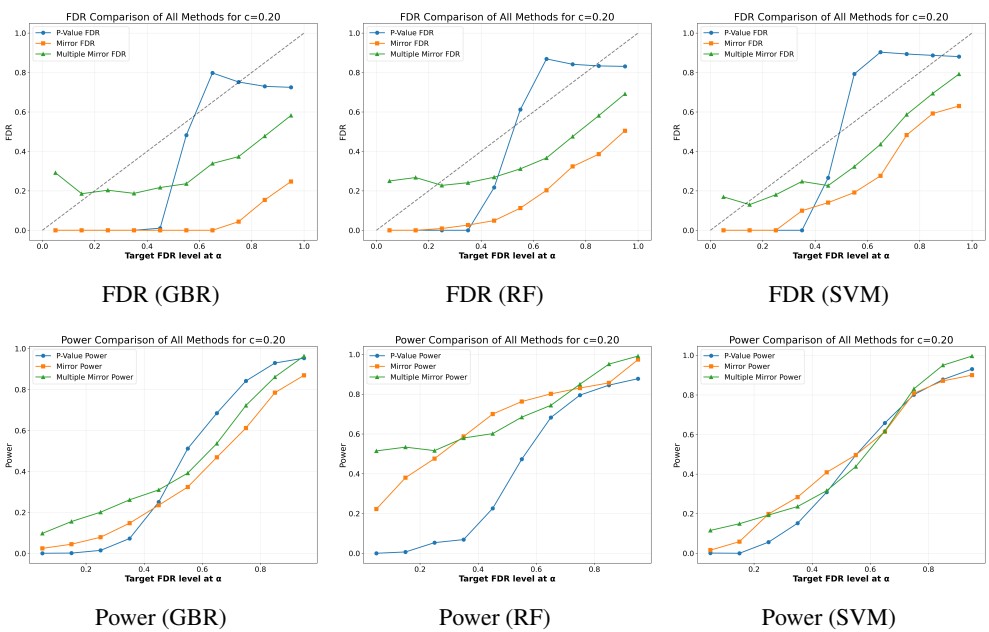

Figure 8: FDR (top) and power (bottom) using GBR, RF, and SVM under Setting (ii) $\sigma(X) = \frac{5.5 - |\mu(X)|}{2}$ at $c = 0.20$.

