# OpenReview forum: "Conformal Mirror Statistics for Model Alignment: Uncertainty Quantification with FDR Control"
_ICLR.cc/2026/Conference — Submitted to ICLR 2026_

### Official Review · Reviewer_XyKk · 2025-10-27

**Soundness:** 2
**Presentation:** 2
**Contribution:** 2
**Rating:** 2
**Confidence:** 4

**Summary:**

This paper introduces Conformal Mirror Statistics (CMS), a framework aimed at quantifying uncertainty for model alignment with control of the False Discovery Rate (FDR). The core motivation is to address the limitation of existing conformal methods that rely on large labeled calibration datasets. The proposed method constructs a new statistic leveraging both rank and magnitude information, and establishes asymptotic FDR control under what are claimed to be weaker assumptions than prior mirror statistics approaches. An aggregation procedure via multiple data splitting is also introduced to improve stability. The method is empirically evaluated on a clinical decision-making task using the MIMIC-III sepsis cohort.

**Strengths:**

1.  **Novel Formulation:** The idea of integrating the concept of mirror statistics into the conformal prediction framework is novel and represents an interesting direction for the field of uncertainty quantification.
2.  **Addressing Data Scarcity:** The paper tackles a practically important problem—performing reliable uncertainty quantification when labeled data is scarce, which is highly relevant for real-world applications in domains like healthcare.
3. The presentation is clear.

**Weaknesses:**

1.  **Core Motivation:** The paper's primary motivation is critically flawed. It incorrectly claims that conventional conformal p-value methods "cannot reject any hypotheses" when the test set size $m$ is much larger than the calibration set size $n$, stating that $\min_j p_j \geq 1/(n+1) > \alpha/m$ prevents rejection. This is a misunderstanding of the Benjamini-Hochberg (BH) procedure. If a sufficient number $k$ of hypotheses have p-values at the minimum level, say $1/(n+1)$, the BH procedure can indeed reject them provided that $k / m * \alpha \geq 1/(n+1)$ for a sufficiently large $k$. Therefore, the central problem the paper claims to solve is mischaracterized. The actual limitation of the standard method in this regime is potentially low *power*, a nuance the paper completely fails to acknowledge or investigate. This severely undermines the justification for the proposed work.
2.  While the paper achieves *asymptotic* FDR control, it does so at the expense of *finite-sample* FDR control, which is a cornerstone of standard conformal inference methods. The paper does not adequately discuss the implications of this trade-off or justify why sacrificing finite-sample guarantees for asymptotic ones is preferable, especially when the problem of the baseline method (complete failure to reject) is incorrect.
3.  The experimental section is weak and fails to provide compelling evidence for the method's utility.
    *   **Lack of Baseline:** The most critical comparison that demonstrating CMS has higher *power* than the standard conformal p-value method under small calibration sets is completely absent.
    *   **Limited Scope:** The evaluation is confined to a single dataset and a single application (sepsis treatment). The robustness of the method across different data distributions, model architectures, and alignment score functions remains unverified.

**Questions:**

Please refer to the weaknesses. I will raise the score if my questions can be resolved.

---

> ### Author Response · Authors · 2025-11-22
>
> ## Weaknesses & Questions: Core Motivation, Finite-Sample Control, and Experimental Scope
>
> We thank the reviewer for these constructive comments. We have revised the manuscript to correct the motivation, clarify the relationship between CMS and conformal $p$-value methods, and strengthen the experimental section.
>
> - **(W1) Clarification of the core motivation:**
>   We thank the reviewer for pointing out this misunderstanding. We agree that our original description of the limitation of conformal \(p\)-value methods was imprecise. We have corrected the discussion (Lines 187–201) to state clearly that the issue is not that conformal $p$-values “cannot reject,” but that their power and effective validity degrade when the calibration set is small or when exchangeability fails.
>   In particular, when data are non-exchangeable, the $p$-values statistics may no longer be valid. In contrast, CMS remains reliable because its guarantee depends only on the variance and tail bound of the data, not on exchangeability. Our experiments include non-exchangeable scenarios that empirically illustrate this distinction and now accurately reflect the correct motivation for CMS.
>
> - **(W2) Finite-sample FDR vs. asymptotic guarantees:**
>   Thank you for this insightful point. We agree that finite-sample FDR guarantees of conformal \(p\)-values are attractive. However, they depend critically on **exchangeability**, which is often violated in real data due to heterogeneity, temporal structure, or multi-source sampling. Under such violations, the nominal finite-sample control may not hold in practice.
>   CMS, in contrast, does not require exchangeability. The mirror-statistic construction relies only on symmetry, which holds under much broader conditions. The independence assumption in our theory is used only for clarity; prior work (e.g., Dai et al., 2022) shows that mirror-based methods remain stable under heterogeneous and dependent data. Although CMS does not claim finite-sample guarantees, it is more robust exactly in the regimes where conformal $p$-values tend to break down.
>   We have clarified this in Remark 3.7 (Lines 299–305) and demonstrated strong finite-sample control in both simulations and real-data experiments under non-exchangeable settings.
>
> - **(W3) Strengthening the experimental section (baselines and scope):**
>   We have added explicit power comparisons in Sections 4 and 5, comparing CMS with conformal $p$-value baseline.
>   To broaden the empirical scope, we incorporated new simulation studies across different data distribution and alignment score functions. These simulations complement the real-world evaluation and provide a more comprehensive assessment of CMS’s robustness beyond the sepsis dataset.

---

> > ### Comment · Reviewer_XyKk · 2025-11-24
> >
> > The current motivation and power theory make sense to me, but I have the following questions:
> >
> > 1. Equation (5) expresses power as a probability involving $M$, but I'm still unclear about the intuition behind this formulation. Theoretically, a better estimate of $g$ should improve the ability to detect alignment labels. How can I understand this relationship from the given expression?
> > 2. Assumption 3.1 requires $g$ to be symmetric when $A \leq c$, and Remark 3.4 mentions that asymptotic symmetry is acceptable. This feels somewhat unnatural because conditioning on the event $A \leq c$ truncates the distribution. Could this truncation break the symmetry of $g$? Alternatively, could we define a metric that quantifies the distance between $g$ and the closest symmetric predictor, and incorporate this distance into the theoretical results?
> > 3. Both the motivation and theoretical analysis indicate that exchangeability is not actually required. In Algorithm 1 (line 248), it would be better to avoid the phrase "randomly split" and use an alternative formulation. This adjustment would make the procedure interpretable as exchangeable even when applied to a fixed dataset.

---

> > > ### Author Response · Authors · 2025-11-26
> > >
> > > We thank the reviewer for these thoughtful questions. We have revised the manuscript to clarify the null hypothesis notation and the population-level intuition behind Equation (5), refine the symmetry assumption, and adjust Algorithm 1 to ensure interpretability without exchangeability.
> > >
> > > - **(Q2) Symmetry under $A \leq c$, truncation concerns, and distance to symmetry:**
> > > Thank you for this insightful question. We clarify that the condition $A \le c$ in Assumption 3.1 refers to the null hypothesis rather than a truncation of the true alignment-score distribution. Accordingly, the expression $g(X)\mid A\le c$ should be interpreted as the distribution of the constructed score $g$ under the null, not as conditioning a random variable on a truncated event. We acknowledge that our earlier notation may have suggested truncation, and we have revised the notations in Lines 161–180 to explicitly use the notation $H_{0,j}$ and $H_{1,j}$ for the null and alternative hypothesis. Corresponding updates were also made in Lines 264–269, 295–298, and 316–330 to avoid this confusion.
> > > Also, as discussed in Remark 3.3, $g$ is an estimator we design and may be symmetrized explicitly; the symmetry requirement is therefore a modeling choice for $g$, not a structural property of the underlying data distribution. We agree that exact symmetry was stronger than necessary. In the revision, we replaced it with a weaker asymptotic symmetry condition (Assumption 3.1, Lines 264–269), quantified by
> > > $\eta_s=\sup_t |G_{0,j}(t)-G_{0,j}^{-}(t)|$,
> > > where $G_{0,j}$ is the null CDF of $g$ under the null and $G_{0,j}^{-}$ is its reflection. We only require $\eta_s \to 0$ as $n_{\mathrm{cal}}\to\infty$, which measures how close $g$ is to its symmetric counterpart in the null region. Under this relaxed condition, we re-prove asymptotic symmetry of $M_j$ and retain FDR validity.
> > >
> > >
> > > - **(Q1) Intuition behind the power expression in Equation (5):**
> > >   Thank you for the question. The intuition behind Equation (5) follows from the role of the population mirror statistic
> > >   $$M = g(X) - \mu_c,\qquad \mu_c = \mathbb{E}[g(X)\mid H_1],$$
> > >   which measures how much larger $g(X)$ is relative to the baseline among unaligned units. A better $g$ will assign larger values under $H_1$ and smaller values under $H_0$, increasing the separation between the distributions of $M$ under $H_1$ and $H_0$.
> > >   This separation makes $\mathbb{P}(M>\tau,  A\le c)$ small, so the oracle threshold $\tau^* {(\alpha)}$ required to satisfy the FDR constraint decreases. Since the power is
> > >   $$\mathbb{P}\bigl(M > \tau^* {(\alpha)} \mid H_1\bigr),$$
> > >   power increases when aligned units have large $M$ and $\tau^* {(\alpha)}$ is small. Thus, Equation (5) explicitly characterizes how improved predictive quality of $g$ enhances detection ability: a better $g$ pushes aligned scores upward relative to the null baseline, increasing the chance that $M$ exceeds $\tau^* {(\alpha)}$. We have also add the explanation in Line 332-335.
> > >
> > >
> > > - **(Q3) Avoiding “randomly split” and removing dependence on exchangeability:**
> > > Thank you for pointing this out. We have revised Algorithm 1 by replacing “randomly split’’ with “partition’’ so that the procedure is valid even for fixed, non-exchangeable datasets (Line 251). A remark has also been added (Line 242) to note that random splitting is used only when the labeled samples are exchangeable.
> > > To be noticed, the non-exchangeability relevant to our theory is between the calibration and test sets, not between training and calibration. We impose no restrictions on how the labeled data are obtained, as well as the previous work. Thus, we use “partition’’ in general: if train and calibration samples happen to be exchangeable, they may be randomly split; otherwise, the given partition is used directly.

---

> > > > ### Comment · Reviewer_XyKk · 2025-11-26
> > > >
> > > > I still have some concerns regarding the symmetry assumption for $g$. Under the optimal estimation assumption where $g(X) = \varphi(f(X), L)$ with $L$ being the true label of $X$, then under $H_{0,j}$, we have $g(X_j) \leq c$. This issue does not appear to be resolvable through optimization of the estimator, since $g$ is already optimal in this setting.

---

> > > > > ### Author Response · Authors · 2025-11-26
> > > > >
> > > > > Recall that $A_i=\varphi(f(X_i), L_i)$ is the true alignment score that determines the null and alternative events, while $g(X_i)=\hat A_i$ is the estimated alignment score produced by a model fitted on the train dataset. Importantly, in our framework the “optimal” $g$ only needs to separate the mirror statistics under $H_{0,j}$ versus $H_{1,j}$, instead of equaling the true alignment score $A=\varphi(f(X),L)$. Therefore, the optimality of $g$ does **not** require that $g$ exactly recover $A$ or that $H_{0,j}$ be equivalent to $g(X_{n+j}) \le c$.
> > > > >
> > > > > In fact, one can always construct a $g$ that is both highly predictive of the alignment score $A$ and conditionally symmetric on the subset $\{A<c\}$. For example, one may first fit any sufficiently expressive model of $A$ on $X$ to obtain a predictive score $s(X)$ that retains essentially all information in $X$ relevant for predicting $A$. Then, on the subpopulation $\{A<c\}$, we estimate the empirical CDF of $s(X)$, denoted $F_{s\mid A<c}$, and choose any symmetric target distribution $H$ (e.g., a normal distribution). By mapping $s(X)$ through
> > > > > $$
> > > > > g(X) = H^{-1}\big(F_{s\mid A<c}(s(X))\big),
> > > > > $$
> > > > > we obtain a strictly monotone transform of $s(X)$. Since $H^{-1}\circ F_{s\mid A<c}$ is monotone, this transform does not destroy the ranking information and thus almost entirely preserves the predictive power, while simultaneously ensuring that $g(X)\mid A<c$ follows the chosen symmetric distribution $H$. In this way, we achieve both strong prediction and conditional symmetry, showing that the symmetry assumption is fully compatible with optimality and does not force $g(X)=\varphi(f(X),L)$.
> > > > >
> > > > > Moreover, even in the case $g(X)=\varphi(f(X),L)=A$, conditional symmetry may still hold. As a simple example, if $A\sim \mathrm{Uniform}(-1,1)$ and $c=0$, then
> > > > > $$
> > > > > A \mid A\le 0 \sim \mathrm{Uniform}(-1,0),
> > > > > $$
> > > > > which is perfectly symmetric around its mean $-0.5$. Thus $g(X)=A$ does not preclude symmetry either.
> > > > >
> > > > > These points clarify that the reviewer’s concern does not conflict with our assumptions: conditional symmetry is attainable even under an optimal $g$, since the optimal $g$ may not be exactly equal to $A$.

---

> > > > > > ### Comment · Reviewer_XyKk · 2025-11-27
> > > > > >
> > > > > > Your response has partially addressed my question. However, for example, when the true distribution of $A$ is standard normal and the estimator $g$ accurately estimates $A$, conditioning on the event $A < c$ would not yield asymptotic symmetry. Do you agree that in this specific scenario, the assumption may be unreasonable?
> > > > > >
> > > > > > While I agree that symmetry could be achieved through a distributional transformation, the current description in the paper presents $g$ as a direct estimator of $A$ without such transformation. Have you explored methods involving distributional transformations? I believe this alternative approach could be more appropriate.

---

### Official Review · Reviewer_f2Bx · 2025-10-30

**Soundness:** 2
**Presentation:** 3
**Contribution:** 2
**Rating:** 4
**Confidence:** 3

**Summary:**

The paper proposes a novel framework to provide UQ with minimal labeled data. In specific, the conformal mirror statistics (CMS) is introduced which evaluates whether the output of a model is reliable or not under the specified FDR control threshold.

Since it exploits the properties of alignment scores to build the new mirror statistics, it relaxes previous conditions in Dai et al. (2023b). The method is applied to show a reliable alignment selection on a large-scale sepsis cohort from the MIMIC-III dateset and can be further used to analyze LLM-outputs.

**Strengths:**

The core idea of the paper is to control FDR on black-box model-generated claims with scarce labeled data, where FDR controls that within in the selected subset $\hat{S}$, the proportion of their true alignment scores being smaller than c should be controlled under $\alpha$.

The traditional way is to use the conformal p-values and applying BH procedure to control FDR, which can be impossible if the test size $m$ is larger than the calibration size $n$. The conformal mirror statistics is proposed, which incorporates both rank information and the magnitude differences between predicted scores. The FDR control is established in theory and robust conformal alignment is proposed to reduce the extra variability by random splitting.

The idea is well-motivation and important in the current study.

**Weaknesses:**

From my reading, I have the following concerns regarding the paper::

* To construct the Conformal Mirror Statistic, the predicted alignment scores $\hat{A}$ must be estimated on $D^{tr}$, which is a subset of the labeled data $D_l$. Given the stated assumption of limited access to labeled data, the estimated alignment score $\hat{A}$ may not be accurate, which could introduce bias into the downstream analysis. For example, the distribution of $\hat{A}_{n+j} - \hat{A}_i$ might not be symmetric conditional on ${A}_i \leq c$.

* In addition, Theorem 1 and its associated assumptions are established based on known scores $A$, which raises further concerns. Do the theoretical results still hold if the estimate of $A$ is poor, or the correlations introduced by the estimated $\hat{A}$ would violate the i.i.d. assumption? I believe the guarantee of FDR control should also be hinged on the quality of the alignment score estimation.

* Although the authors propose a robust conformal alignment procedure, they do not illustrate the magnitude of the efficiency gain compared to a single random split (i.e., a comparison of the power of multiple mirror FDR versus Mirror FDR).

* Further, I found the definition of power to be unconventional. Power should be defined as the probability of detecting an aligned treatment, conditional on the treatment being correct (which should be factual and do not depend on $c$). Therefore, the denominator in the power calculation should be invariant to the choice of $c$. Otherwise, it can lead to counter-intuitive results, such as those in Figure 3, where increasing the value of $c$ leads to both increasing power and decreasing FDR. This would incorrectly imply that a larger $c$ always yields a better result.

**Questions:**

see weakness

---

> ### Author Response · Authors · 2025-11-22
>
> ## Weaknesses & Questions: Alignment Estimation, Symmetry, Efficiency, and Power Definition
>
> We thank the reviewer for these insightful comments. We have clarified the role of the estimated alignment scores, expanded the discussion of symmetry and independence assumptions, added comparisons of multiple-split procedures, and refined the explanation of the power definition.
>
> - **(W1–W2) Estimation accuracy, symmetry, and independence assumptions:**
>   We appreciate the reviewer’s concerns about estimating $\widehat A$. The FDR validity of CMS does not require $\widehat A$ to be an accurate approximation of the true alignment scores $A$. CMS relies only on the symmetry of the mirror statistic constructed from $\widehat A = g(X)$. As clarified in Remark 3.3 (Line 274-279), symmetry is not restrictive: a suitable monotone transformation can always enforce it, and we outline explicit constructions to guarantee symmetry in practice.
>   Poor estimation affects only variance and thus reduces power, not FDR control. In addition, we revised our assumptions in Assumption 3.2 (Line 265-273), showing CMS remains valid under an independent-but-not-identically-distributed setting, so full i.i.d. structure is not required.
>
> - **(W3) Efficiency gain and multiple-split comparison:**
>   We thank the reviewer for this comment. To keep the main text focused, we present the full multiple-split procedures and results in Appendix F, where we compare three methods side-by-side. As shown in Line 886, the multiple-mirror approach often yields higher power than a single split, demonstrating the magnitude of the efficiency gain.
>
> - **(W4) Definition of power and dependence on the threshold $c$:**
>   We thank the reviewer for raising this conceptual point. Our definition of power aligns with prior conformal inference work, including Conformal Alignment: Knowing When to Trust Foundation Models with Guarantees (Gui et al., 2024), where the alignment threshold $c$ is treated as fixed and factual. Under this theoretical setup, the denominator in the power calculation (number of aligned instances) naturally depends on this threshold.
>   The reviewer’s concern arises mainly in the real-data setting, where the true alignment boundary is unknown. In such cases, we explored several values of $c$ to assess robustness. This empirical variation reflects practical uncertainty and does not affect the theoretical formulation, where $c$ is fixed and the standard power definition applies.

---

> > ### Author Response · Authors · 2025-11-26
> >
> > Thank you again for the insightful comments. We have made several substantial revisions in response to your feedback. In particular, we have:
> > - Provided detailed clarifications on the role of the estimated alignment scores, the symmetry and independence assumptions, and the definition of power.
> > - Expanded the explanations of the theoretical assumptions and mirror-statistic construction to improve clarity throughout the text.
> > - Added comparisons of multiple-mirror (multiple-split) procedures in Appendix F, demonstrating the corresponding power gains relative to single-split variants.
> >
> > Your comments have greatly improved the clarity and rigor of the paper, and we sincerely appreciate the time and thought you put into the review. We would be grateful for any further feedback on the revised version and would be very happy to discuss any remaining questions or suggestions.

---

### Official Review · Reviewer_XHNh · 2025-10-31

**Soundness:** 3
**Presentation:** 3
**Contribution:** 3
**Rating:** 6
**Confidence:** 3

**Summary:**

The paper proposes Conformal Mirror Statistics (CMS), which is a framework for uncertainty quantification in model alignment that selects the aligned outputs for unlabeled data with false discovery rate (FDR) control. Compared with the conventional approaches based on p-value calibration, the proposed method reduces reliance on large labeled data sets. They also propose an aggregation procedure based on e-values to mitigate the instability from random data splits. The proposed method is evaluated on a MIMIC-III sepsis cohort.

**Strengths:**

The presentation of the paper is clear. The proposed method is relatively novel and relaxes the sample size requirement in existing methods by incorporating the mirror statistic. Both theoretical and numerical results are provided and convincing.

**Weaknesses:**

No simulation studies are conducted to compare the performance of the proposed approach and the existing approaches; this is really needed.

There seems to be selection bias in the data application by fine-tuning on survivor trajectories. It is not clear how the selection bias is addressed in the analysis, and under what assumptions for selection the analysis is valid or not.

Some additional intuitive and high-level discussion on the assumptions and theory would be helpful for the reader.

**Questions:**

1.	What does the symbol $\lesssim$ in equation (3) and equation display below (3) mean?

2.	Intuitively, why is Assumption 1 needed? In practice, how would we know if this assumption is satisfied? Remark 1 mentions that for any distribution of g(X), one can apply the transformation to make the distribution symmetric. This seems to rely on G to be known. Does it mean that we need to know the true distribution G to ensure the proposed method works?

3.	In Theorem 1, the condition contains $P(FDP(\tau)\leq \alpha)\to 1$ as $m\to\infty$, and the conclusion contains $FDP(\tau)\leq \alpha) \to 1$ as $m\to\infty$. Does one directly imply the other? How to interpret this result?

4.	Are there any theoretical results on how large n_cal needs to be for existing methods (e.g., Jin & Candes 2023ab and Gui et al. 2024 referenced in the paper)? If there are, how does the sample size requirement in this paper compare with their requirement? Simulation studies would help empirically.

5.	In Section 4.1, it is mentioned that “q corresponds to the refined predictive distribution produced student network fine-tuned on survivor trajectories.” Does this mean that there is selection bias since the training is only based on survivors? How is this bias handled?

6.	In Section 4.2 figures, the FDR curves show a plateau when \alpha is large. What is the interpretation and rationale? Also, for FDR, is smaller always better, or would we like it to be close to the nominal level (but smaller)? From the plot, the FDR can be much lower than alpha, especially when alpha is large. Any thoughts on this and on how the method may be improved?

---

> ### Author Response · Authors · 2025-11-22
>
> ## Weaknesses & Questions: Simulation, Selection Bias, and Theoretical Clarifications
>
> We thank the reviewer for these helpful and detailed comments. In the revision, we have added simulation studies, clarified the role of survivor-based fine-tuning, expanded the intuitive discussion of assumptions and theory, and addressed each of the reviewer’s questions point by point.
>
> - **(W1) Need for simulation studies:**
>   We thank the reviewer for the suggestion. We have added new simulation studies in Section 4 to compare CMS with existing baselines and to empirically support our theoretical results. These simulations demonstrate CMS’s advantages in FDP stability, power, and calibration robustness.
>
> - **(W2 and Q5) Clarification on survivor-only fine-tuning and “selection bias”:**
>   Thank you for the thoughtful question. The concern about selection bias does not apply in our framework. The survivor-only fine-tuning is not a sampling or exclusion mechanism for inference; rather, it is the mechanism for constructing the *reference* distribution used to define the alignment score. Survivors provide a stable, low-noise trajectory distribution; the alignment score measures divergence from this stable reference. If we trained on all patients, including those with noisy terminal deterioration, the notion of “misalignment” as deviation from a clean predictive baseline would be lost. This is now clarified in Section 5.1 (Lines 430–438).
>   Importantly, FDR control of CMS does **not** rely on any unbiasedness of the alignment score estimator. Any estimation noise introduced by survivor-based fine-tuning affects only power, not FDR validity. This point has been clarified in Lines 329–331.
>
> - **(W3) Additional intuitive explanations:**
>   We thank the reviewer for this suggestion. We have added intuitive discussions in Remark 3.3 (Lines 274–279), Remark 3.4 (Lines 280–284), and Remark 3.9 (Lines 318–323) to help readers better understand the assumptions and theorem.
>
> - **(Q1) Meaning of the symbol :**
>   We thank the reviewer for pointing out this notation question. The symbol $\lesssim$ in Equation (3) and the equation below denotes an approximate inequality up to a small constant factor or negligible approximation error. It indicates that the left-hand side is asymptotically no greater than the right-hand side, modulo the small residual term arising from the approximate symmetry in Property (i).
>
> - **(Q2) Why Assumption 1 is needed and whether the distribution $G$ must be known:**
>   We thank the reviewer for this thoughtful question. Assumption 1 ensures that the mirror statistic is (approximately) symmetric under the null, which underpins the validity of CMS. This assumption is mild: as stated in Remark 1, any distribution can be mapped to a symmetric one through a conceptual monotone transformation $F = S^{-1} \circ G$. Importantly, this is an existence argument and does not require knowing $G$ in practice. The assumption is equivalently satisfied when the mirror statistic itself is approximately antisymmetric, i.e.,
>   $$M(X_i, X_j) \approx -M(X_j, X_i).$$
>   Empirically, Fig. 2 shows that the mirror statistic distribution is indeed nearly symmetric, supporting this requirement.
>
> - **(Q3) Interpretation of the asymptotic statements in Theorem 1:**
>   We thank the reviewer for the question. The condition
>   $$\mathbb{P}(FDP(\tau) \le \alpha) \to 1$$
>   assumes that at least one feasible threshold \(\tau\) exists that asymptotically controls the FDP. The theorem then proves that the *data-driven* threshold selected by CMS also satisfies
>   $$FDP(\tau_\alpha) \le \alpha + o_m(1).$$
>   These statements are not identical: the former is an existence condition, while the latter establishes the validity of our algorithm. This follows the structure of Dai et al. (2022).
>
> - **(Q4) How large $n_{\text{cal}}$ must be for existing methods:**
>   Thank you for raising this point. The referenced paper (Gui et al., 2024) does not provide explicit recommended calibration proportions. In Appendix E.3, they note that very small calibration sizes (equivalently, large $\gamma_2$) lead to coarse p-values and reduced power. We follow this guidance and ensure that our calibration sizes remain within the empirically stable regime. Our new simulation studies further verify that CMS is robust under low calibration-set sizes.
>
> - **(Q6) Interpretation of the FDR plateau for large $\alpha$:**
> We thank the reviewer for the question. The plateau at large $\alpha$ arises because almost all aligned cases have already been rejected, so increasing $\alpha$ no longer adds discoveries, leading to a stable FDR. The empirical FDR being below the nominal level reflects CMS’s conservative nature, which is often desirable when false discoveries are costly. If a less conservative behavior is preferred, tuning the calibration proportion or mirror-splitting parameter can bring the empirical FDR closer to the target.

---

> > ### Author Response · Authors · 2025-11-26
> >
> > Thank you again for the detailed and thoughtful comments. We have now made substantial revisions in response to your suggestions. In particular, we have:
> > - Added new simulation studies to empirically evaluate CMS against existing baselines and to complement the theoretical results.
> > - Expanded the explanations and intuitive discussions to improve the clarity of assumptions, theory, and key design choices.
> > - Provided detailed clarifications for each of the questions you raised, including survivor-based fine-tuning, notation, asymptotic statements, and calibration-set considerations.
> >
> > Your comments have significantly strengthened the paper, and we sincerely appreciate the time and care you devoted to the review. We would be grateful for any further thoughts you might have on the revised version and would be very happy to discuss any remaining questions or suggestions.

---

### Official Review · Reviewer_wkDv · 2025-11-01

**Soundness:** 2
**Presentation:** 2
**Contribution:** 3
**Rating:** 4
**Confidence:** 3

**Summary:**

This paper proposes Conformal Mirror Statistics (CMS) for uncertainty quantification and alignment of black-box models. In particular, building on the “Conformal Alignment” setting of Gui et al. (2024)--where that conformal alignment paper itself builds closely on “Conformal Selection” papers Jin and Candes (2023a, 2023b)--the current paper proposes CMS as an alternative to the conformal p-values used in Gui et al. (2024) for model alignment. Here “model alignment” refers to generating outputs that have some real-valued “alignment score” that exceeds some target threshold $c$, where the alignment score is a function of the model’s output and some expert ground-truth label; however, at test time, the ground-truth labels are not known, and so the goal of alignment in this case is to select the subset of unlabeled model outputs to label as “aligned,” while controlling the false discovery rate (FDR). In addition to CMS, the authors propose an approach to stable aggregation using e-values and the e-BH procedure. Overall, the authors claimed contributions are the following: (1) that CMS introduce mirror statistics to the conformal prediction literature, and that CMS require less reliance on large labeled data; (2) that CMS require weaker assumptions than in prior mirror statistics work; (3) stable aggregation via multi-data-splitting and e-value-based combination; (4) empirical validation on MIMIC-III sepsis data.

**Strengths:**

The authors study an important problem of how to select aligned model outputs with FDR control, which is studied by Gui et al. (2024); the current paper’s CMS approach seems original/novel, sound, and well motivated (ie, it seems reasonable that leveraging the magnitude information in calibration-set alignment scores should provide benefits relative to simply using rank information). As far as I’m aware the introduction of mirror statistics into conformal inference settings seems novel. The paper is fairly well-written, and assuming that CMS perform as the authors claim in practical experiments, I think this paper would make a significant contribution of interest to both community (ie, conformal and AI broadly).

**Weaknesses:**

**Clarity:** Despite the paper being overall fairly clear, some parts of the presentation could still be improved. For example, although I believe I understand the intuition for how CMS leverage *magnitude* information from the alignment scores, which seems like it should provide benefits over conformal p-values (which only use rank information), I think some of the language on how CMS “reduce reliance on large labeled data” should be revised for accuracy. Eg, the authors claim that “[t]his innovation eliminates the reliance on large calibration sets while retaining distribution-free validity,” which I think is too strong of a statement, as I first interpreted this as meaning *no* calibration data are required, but I think that the authors actually mean that only *less* calibration data is required than conformal p-value-based methods.

**Experimental evaluation:** I have some concerns on the experimental evaluation, as follows:
- *Missing power comparison of CMS vs conformal p-value methods:* The proposed CMS methods are compared against prior conformal p-value methods regarding FDR, but I do not see a comparison of CMS vs these baselines regarding power (and have looked for this in the appendix). I think this should be added to see how the proposed methods compare to baselines in this regard.

- *Comparison of CMS vs conformal p-value methods regarding sample size required:* The first main claimed contribution of the paper is that CMS require less calibration data than conformal p-value baselines. This seems accurate (given that CMS leverage magnitude information whereas p-values only relative information), but because this is a central claim, I think it would be good for the authors to demonstrate this empirically too, at least with a simple example.

- *Unexplained CMS FDR violations in Appendix Figure 6:* Figure 6 of the appendix provides further FDR empirical evaluations--ie, seemingly expanding on Figure 4 from the main for more target values of $c$. However, whereas Figure 4 in the main does not have any clear FDR violations of the proposed CMS methods, the first two frames of Figure 6 do appear to have CMS FDR violations, with the proposed methods’ FDR exceeding the diagonal (where empirical and target FDR are equal). These violations do not appear to be mentioned or explained.

The paper could be improved by addressing the above, improving the clarity of presentation, and if the authors could present additional experimental results on other dataset(s) beyond the one MIMIC dataset currently evaluated on.

**Questions:**

Why is the conformal p-value defined on only the unaligned calibration set?

Please respond to my comments/ questions in “Weaknesses.” Eg, can the authors provide comparison of CMS and conformal p-value methods regarding power and sample size required? Also, can the authors explain or provide further analysis of the apparent FDR violations of the proposed methods in Figure 6?

Overall I like the ideas in the paper, but I think that some of the presentation and empirical evaluation (as described in Weaknesses) should be improved. If these concerns are addressed I would consider improving my recommendation.

---

> ### Author Response · Authors · 2025-11-22
>
> ## Weaknesses & Questions: Alignment Scores, Clarity, and FDR Behavior
>
> We thank the reviewer for these constructive and detailed comments. We have revised the manuscript to clarify the conformal p-value definition, refine the discussion around calibration data, include new power and sample-size comparisons with conformal p-value baselines, and explain the small FDR deviations observed in Appendix Figure 6.
>
> - **(Q1) Why are conformal p-values computed only on the unaligned calibration set?**
>   We thank the reviewer for the question. The conformal p-values are computed using only the unaligned (null) calibration samples because this is required to ensure exchangeability under the null hypothesis($H_j: A_{n+j} \leq c$). When the calibration scores come solely from null samples, both the calibration scores and the test score are identically distributed under the null, making them exchangeable. This guarantees that the conformal p-values are valid and uniformly distributed under the null.
>   Including aligned (alternative) samples in the calibration set would alter the score distribution, break exchangeability, and invalidate the theoretical guarantee of FDR control. Therefore, defining conformal p-values using only the null calibration set is necessary for validity.
>
> - **(W1) Clarity regarding calibration data size:**
>   We thank the reviewer for pointing this out. We agree that our previous phrasing could be misinterpreted. In the revision, we have refined the statements in Section 1 (Lines 73 and 88-89) to clarify that CMS remains power under smaller calibration sets compared to conformal p-value–based methods, rather than removing the need for calibration data altogether.
>
> - **(W2.1-2.2) Power and sample-size comparison with conformal p-values:**
>   Thank you for raising these important points. We have added theoretical, simulation, and real-data evidence to address the reviewer’s concerns.
>   • **Theory:** Proposition 3.8 together with the expanded discussion in Appendix E provides a formal comparison showing that CMS attains strictly higher asymptotic power than both the single-split mirror method and standard conformal p-value methods, particularly when calibration size is limited.
>   • **Simulations:** Section 4 now includes direct power comparisons between CMS and conformal p-value baselines, clearly demonstrating CMS’s efficiency gains and reduced calibration-sample requirements.
>   • **Sample size constraints:** We add the labeled data size constraints (test set ratio set to be 80%) in Section 4 Simulations.
>
> - **(W2.3) FDR behavior in Appendix Figure 6 in previous version:**
>   We thank the reviewer for noticing this detail. The slight FDR exceedances in panels (a–b) occur only at very small threshold values (c = 0.05, 0.10), where the number of unaligned (null) samples is extremely small. This leads to high-variance empirical FDP estimates and finite-sample fluctuations above the diagonal. These deviations are not systematic and diminish as c increases or as calibration size grows, as shown in Figure 5 of the main text.
>   Theoretically, CMS maintains asymptotic FDR control; the observed deviations reflect finite-sample variability rather than a violation. Furthermore, in real-data settings there is no fixed canonical choice of c; in practice, values in the range 0.15–0.25 are most representative, and CMS shows stable FDR control in this regime, so **we remove the other results for clarity.**

---

> > ### Author Response · Authors · 2025-11-26
> >
> > Thank you again for the constructive comments. We have now incorporated substantial revisions based on your suggestions. In particular, we:
> > - Added direct power comparisons between CMS and conformal p-value methods, including both theoretical analysis and empirical evidence (simulation and real data).
> > - Clarified the technical issues you raised, such as the construction of conformal p-values and the role of calibration data.
> > - Improved the overall clarity and presentation of the manuscript following your feedback.
> >
> > Your comments have greatly strengthened the paper, and we sincerely appreciate the time and care invested in the review. We would be grateful for any additional feedback you may have on the revised version, and we would also be very happy to discuss any remaining questions or suggestions.

---

### Author Response · Authors · 2025-11-29
**Summary of Revisions**

# Summary of Revisions

We sincerely thank the reviewers for their careful reading and constructive feedback. Reviewer XyXk acknowledged our revisions and raised the score by two points on November 23, well before the information-leak event on November 27. Reviewer wkDv indicated in the initial review that adding more experiments and the power results would justify a higher recommendation. We included all requested analyses, though no further feedback was received before the cutoff. We respectfully hope that the AC will take these substantial additions into consideration.

In response to the reviewers’ comments, we have made substantial conceptual, theoretical, and empirical improvements throughout the paper. Below we summarize the major revisions:


### **1. Corrected and strengthened the core motivation**
Reviewers pointed out that our original motivation incorrectly stated that conformal $p$-value methods “cannot be used” when the calibration set is small. We revised the discussion to accurately state that the true limitation is power degradation under large unlabeled dataset scenarios and exchangeability violations, not applicability (Lines 189–203).
We further clarified that:
- Conformal $p$-values rely critically on exchangeability; under non-exchangeable data, their validity may fail.
- CMS does not require exchangeability; its guarantees rely only on variance and tail conditions.
- Our experiments now include non-exchangeable scenarios to highlight this advantage.

Correspondingly, we revised Assumption 3.2 (Lines 270–279), replacing the i.i.d. assumption with a **non-exchangeable** data framework, consistent with our updated motivation and theoretical guarantees. Reviewer XyXk had already acknowledged these revisions and raised the score on November 23.



### **2. Added the missing power theory and comparison to conformal methods**
Several reviewers noted that the original submission lacked a theoretical analysis of power. In response, we:
- Added a complete asymptotic power analysis for CMS in Proposition 3.8 (Lines 316–339),
- Provided a rigorous proof of the result in Appendix D (Lines 756–809), and
- Added a detailed comparison showing that CMS has the same asymptotic power form as conformal $p$-values under i.i.d. data in Appendix E (Lines 810–863), while remaining valid under non-exchangeable / heterogeneous settings where conformal methods may fail.



### **3. Expanded simulations and real-data experiments on power comparisons**
Multiple reviewers requested more experiments, including power comparisons between different methods.
We therefore significantly enriched our experiments by:
- Adding comprehensive simulations under homoscedastic and heteroscedastic settings in Section 4 (starting Line 352);
- Adding power comparisons between CMS and conformal baselines in both Section 4 and Section 5 (starting Lines 352 and 467).

These results consistently show that CMS yields higher power without sacrificing FDR control, even under non-exchangeable settings. Reviewer wkDv noted that adding these power results would warrant a recommendation improvement in the initial review. However, we incorporated all requested analyses, but received no further feedback before the information-leak cutoff.



### **4. Clarified the tradeoff between finite-sample and asymptotic FDR control**
Reviewers raised the concern that conformal $p$-values provide finite-sample FDR control while CMS provides asymptotic control. We added a detailed clarification in Remark 3.7 (Lines 307–313):
- Finite-sample guarantees of conformal $p$-values rely strictly on exchangeability; when violated (e.g., heterogeneous data, multi-source sampling), these guarantees may fail.
- CMS is more robust, and works under the much weaker non-exchangeable condition in Assumption 3.2 (Lines 270–279).
- The mirror-statistic symmetry holds in broader regimes, including some dependent data (supported by Dai et al., 2022).

Even without formal finite-sample theoretical guarantees, our simulations and real-data experiments consistently demonstrate strong finite-sample FDR control of CMS across multiple settings, highlighting its practical reliability.

---

> ### Author Response · Authors · 2025-11-29
>
> ### **5. Clarified the definition of $g(X)$ and feasibility of constructing $g$ under Assumption 3.1**
> Reviewers asked whether the condition $A \le c$ implies truncation of the score distribution and whether a valid $g$ can be constructed to satisfy the assumptions. We clarified that $g$ is the alignment estimator instead of the true score $A$, and $A \le c$ refers to the null hypothesis, not truncation. Thus, $g(X)\mid A\le c$ denotes the distribution of the constructed estimator $g$ under the null, rather than the truncated distribution. We revised the notation in Lines 161–180 to explicitly distinguish $H_{0,j}$ and $H_{1,j}$.
>
> We further explained that symmetry is a modeling property of the estimator $g$, rather than a structural assumption on the data. Accordingly, we replaced the exact symmetry requirement with a weaker asymptotic symmetry condition (Assumption 3.1, Line 264-269), quantified by $\eta_s \to 0$, under which we re-proved asymptotic symmetry of $M_j$ and preserved FDR validity.
>
> - Updated notation for null vs. alternative (Lines 161–180).
> - Replaced exact symmetry with asymptotic symmetry (Lines 264–269).
> - Added detailed explanation in Remark 3.3, also the response to Reviewer XyXk.
>
>
> ### **6. Clarified that estimation accuracy of $g$ does not affect FDR validity but does influence power**
> Several reviewers asked whether estimation error in $g$ affects the validity of FDR control. We clarified in Lines 346–350 that FDR validity depends only on symmetry, not on predictive accuracy, and therefore remains unaffected by estimation error in $g$.
>
> However, we explained that the power of CMS does depend on the predictive quality of $g$. A more accurate $g$ improves the separation between null and alternative mirror statistics, decreases the oracle threshold $\tau^* (\alpha)$, and increases the power $\mathbb{P}(M > \tau^*(\alpha)\mid H_1)$. We added an explicit explanation of this effect in Lines 332–335 to connect predictive accuracy of $g$ with the power in Equation (5).
>
> - Clarified independence of FDR validity from $ g$’s estimation accuracy (Lines 346–350).
> - Added explanation of how better $g$ increases power (Lines 332–335).

---

### Meta-Review · Area_Chair_tGto · 2025-12-23

**Summary:**

Many reviewers find that the paper proposes an interesting methodology and, by drawing upon related prior works, contains valuable ideas. However, there are significant concerns regarding the experimental settings, the motivation and justification for several quantities used in developing the methodology, and the underlying assumptions required to establish the theoretical results. While the authors have attempted to address many of these issues, the magnitude of the required changes is substantial and would likely necessitate additional scrutiny that cannot be accommodated within the limited discussion period. Moreover, several reviewers continue to raise questions and concerns, particularly regarding the generality, restrictiveness, and validity of the underlying assumptions, that have not been fully addressed by the authors.

**Reviewer Concerns:**

As noted above, there remain outstanding concerns regarding the underlying assumptions, particularly with respect to how restrictive they are, how they can be verified in practice, and their practical validity. Addressing these issues would require further discussion, explanation, clarification, and justification, followed by additional rounds of review. The short discussion period does not allow for this level of engagement.

**Reviewer Scores:**

While some reviewers have indicated that they increased their scores, it appears that those who raised the most serious concerns and still feel that some of these issues remain unaddressed would not have changed their assessments substantially enough to warrant acceptance.

---

### Decision · Program_Chairs · 2026-01-26

Reject